

# Large-scale subsidence promotes convection in sub-Arctic mixed-phase stratocumulus via enhanced below-cloud rain evaporation

Gillian Young[1], Paul J. Connolly[1], Christopher Dearden[1], and Thomas W. Choularton[1]

[1]Centre for Atmospheric Science, School of Earth and Environmental Sciences, University of Manchester, Manchester, UK.

*Correspondence to:* Thomas W. Choularton (choularton@manchester.ac.uk)

**Abstract.**

Large-scale subsidence, associated with high pressure systems, is often imposed in large-eddy simulation (LES) models to maintain the height of boundary layer (BL) clouds. Previous studies have considered the influence of subsidence on warm, liquid clouds in subtropical regions; however, the relationship between subsidence and microphysics has not specifically been studied, especially in mixed-phase clouds. For the first time, we investigate how widespread subsidence associated with synoptic-scale meteorological features can affect the microphysics of sub-Arctic marine mixed-phase stratocumulus (Sc) clouds. Modelled with LES, four idealised scenarios – a stable Sc, varied droplet ($N_{drop}$) or ice ($N_{ice}$) number concentrations, and a warming surface – were subjected to different levels of subsidence to investigate the cloud microphysical response. We find strong microphysical sensitivities to large-scale subsidence, indicating that high pressure systems in the ocean-exposed low-, or sub-, Arctic regions have the potential to generate turbulence and changes in cloud microphysics in any resident BL mixed-phase clouds.

Increased convection is modelled within the clouds with increased subsidence, driven by radiative cooling at cloud top and rain evaporative cooling below cloud base. Subsidence strengthens the BL temperature inversion, reducing entrainment and allowing the liquid- and ice-water paths (LWP, IWP) to increase. Through increased cloud top radiative cooling and subsequent convective overturning, precipitation production is enhanced: rain particle number concentrations ($N_{rain}$), in-cloud production rates, and below-cloud evaporation rates increase with increased subsidence. In these liquid-dominated mixed-phase clouds, subsidence contributes towards increased BL inversion strength, BL turbulent kinetic energy (TKE), and cloud LWP. Ice number concentrations, $N_{ice}$, play an important role, as greater concentrations suppress the liquid phase; therefore, $N_{ice}$ acts to mediate the strength of turbulent overturning induced by subsidence and longwave radiative cooling in the modelled mixed-phase clouds. With a warming surface, a lack of – or low – subsidence allows for rapid BL TKE coupling, leading to a heterogeneous cloud layer, cloud top ascent, and cumuli formation below the Sc cloud. In these scenarios, higher levels of subsidence act to stabilise the Sc layer: the combination of these two forcings counteract one another to produce a stable, yet dynamic, Sc layer.



# 1 Introduction

Arctic mixed-phase clouds are long-lived, and widespread single-layer stratocumulus (Sc) decks are common in the autumn, winter, and spring. These stable Sc clouds are maintained and driven by convection induced by strong radiative cooling at the boundary layer (BL) inversion (e.g. Feingold et al., 2010; Morrison et al., 2012). In numerical models, mechanisms affecting

the break up of these Sc clouds – including glaciation (e.g. Harrington et al., 1999; Prenni et al., 2007; Young et al., 2017) or break up into convective cumulus, as occurs in cold-air outbreaks (CAOs) – are often too efficient, leading to inaccurate radiative predictions in the sub-Arctic region.

Several studies (e.g. Harrington et al., 1999; Harrington and Olsson, 2001; Prenni et al., 2007; Morrison et al., 2012; de Boer et al., 2011; Young et al., 2017) have addressed the issue of premature dissipation of modelled mixed-phase Sc through

glaciation, often concluding that the cause is an over-active ice phase and strong influence of the Wegener-Bergeron-Findeisen (WBF) mechanism. The WBF mechanism influences a constantly changing, unstable microphysical structure which often causes cloud glaciation when modelled as the represented processes are too efficient. Whilst these clouds are microphysically unstable, they can persist for long periods of time; periods during which they may move geographically.

In a CAO, these widespread Sc decks are transported southwards from over the sea ice to over the warm ocean. These

clouds often display closed cellular structure at first, where narrow downdraught rings surround broad updraught columns to produce a cloud state which is highly reflective to incoming solar radiation (Schröter et al., 2005; Feingold et al., 2010). Similar to warm marine Sc (Kazil et al., 2014), little precipitation reaches the surface in regions of closed cellular convection. With motion southwards, increased sensible heat fluxes and BL depth (Young et al., 2016) promote the development of precipitation through induced cloud turbulence and convection (Müller and Chlond, 1996; Wang and Feingold, 2009a; Feingold et al., 2010;

Kazil et al., 2014). Observations in tropical regions show that the majority of rain produced in regions of closed cellular convection evaporates below cloud (Wood et al., 2011); evaporation which has been previously shown to be instrumental in modelling the transition to open cell cumulus (Savic-Jovcic and Stevens, 2008).

Transitions between closed and open cellular convection have been the focus of several studies (e.g. Wang and Feingold, 2009b; Feingold et al., 2010; Wood et al., 2011). Cleaner scenarios (with lower aerosol particle and cloud droplet number

concentrations) have been found to be susceptible to the formation of open cells due to the resultant larger droplet sizes – through the aerosol-indirect effect – which participate efficiently in collision-coalescence interactions to form precipitation (Feingold et al., 2010; Wang and Feingold, 2009a; Rosenfeld et al., 2012). Studies focusing on open cells have often reached the same conclusion that precipitation plays a key role in their development (e.g. Wang and Feingold, 2009a; Feingold et al., 2010; Wood et al., 2011; Rosenfeld et al., 2012). However, in subtropical marine Sc, the development of drizzle has been

found to be influenced more by larger-scale meteorology, such as moisture fluxes and temperature fluctuations, than aerosol-cloud interactions (Wang et al., 2010). Thermodynamic interactions, namely diabatic processes such as latent heat release from condensation and cloud top radiative cooling, have been shown to strongly influence the broadening of convective cells in CAOs (Müller and Chlond, 1996; Schröter et al., 2005). Such interactions are thought to have an important role in generating



dynamical overturning in both the stable mixed-phase Sc upstream and the closed-to-open cellular transitions downstream in CAOs.

BL depth grows to its maximum extent at the peak of a CAO (Fletcher et al., 2016). The BL is able to grow more freely with a weaker inversion, and inversion strength can typically be related to large-scale subsidence (Myers and Norris, 2013). With less subsidence, the velocity of entrained air increases due to a greater cloud top (CT), and BL, height (van der Dussen et al., 2016). Sandu and Stevens (2011) showed that decreasing the imposed large-scale subsidence in a large-eddy simulation (LES) model slows the transition from Sc to cumulus when considering warm, liquid only clouds, as the Sc-topped layer is sustained for longer under lower levels of subsidence. As a result, the authors found that a thicker, more homogeneous Sc layer could be modelled when a lower large-scale subsidence imposed. Furthermore, recent modelling studies indicate that less subsidence extends the lifetime of sub-tropical marine Sc over a warming ocean surface and allows the liquid-water path (LWP) to increase in the absence of precipitation (van der Dussen et al., 2016). These findings suggest that synoptic-scale meteorological features have the potential to influence the microphysical evolution of marine mixed-phase Sc.

High pressure systems are associated with large-scale subsidence and, in turn, strong BL inversions (Myers and Norris, 2013). In the Arctic (over sea ice), high surface pressure anomalies – associated with anti-cyclonic circulation patterns – have been shown to produce less cloudy BLs (Stramler et al., 2011; Morrison et al., 2012) with comparison to cyclonic circulation patterns (Kay and Gettelman, 2009). However, in marine environments, these subsidence-enforced BL inversions lead to increased cloudiness and mixing within the BL whilst maintaining a shallow depth (Myers and Norris, 2013). Regions of high surface pressure are often found upstream of CAOs in the Arctic (Fletcher et al., 2016). Walsh et al. (2001) found that European CAOs could be linked to a negative North Atlantic Ocean (NAO) index, and positive biases in the sub-Arctic mean sea level pressure, by considering CAO climatologies over the period 1948-99. These high surface pressure anomalies are thought to be instrumental to the formation of the CAO flows (Kolstad et al., 2009). With a negative NAO index, high pressure dominates in the European Arctic, moist westerly air flows are weakened, and cold air is able to move southwards towards the European continent more easily.

The role of subsidence in CAO cellular transitions has been suggested in previous studies (e.g. Müller and Chlond, 1996; Feingold et al., 2015); however, its influence on the microphysical evolution of the modelled mixed-phase clouds has not been scrutinised in detail. Here, we investigate the effect subsidence has on the microphysical stability of mixed-phase Sc clouds by considering both idealised test scenarios susceptible to the formation of precipitation and more realistic scenarios with a warming oceanic surface.

## 2  Methods

### 2.1  Model setup

The influence of large-scale subsidence on marine mixed-phase Sc cloud microphysics is investigated using the UK Met Office Large Eddy Model (LEM, UK Met Office, Gray et al., 2001). The set up is the same as that used by Young et al. (2017), whose study gives further details on the model itself. The Piacsek-Williams (PW, Piacsek and Williams, 1970) centred differ-



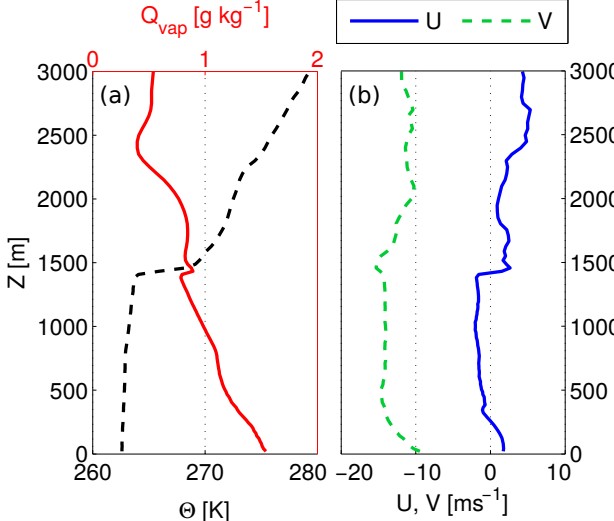

**Figure 1.** Profiles of potential temperature ($\Theta$), water vapour mixing ratio ($Q_{vap}$), and wind speed (U, V) used to initialise the LEM.

ence scheme is used for momentum advection – conserving linear and quadratic terms to good accuracy, allowing for energy conservation (Piacsek and Williams, 1970; Gray et al., 2001) – whilst the total variation diminishing (TVD) monotonicity-preserving scheme of Leonard et al. (1993), known as ULTIMATE, is used for scalar advection (Gray et al., 2001; Shipway and Hill, 2012).

Cyclical boundary conditions and a damping layer (500 m below model lid) were imposed. Vertical profiles of potential temperature ($\Theta$), water vapour mixing ratio ($Q_{vap}$), and wind speed (U and V) were implemented to initialise the model (Fig. 1): these profiles were extracted from previous LEM runs of Arctic mixed-phase Sc, specifically from 10 h in the ACC ocean case detailed by Young et al. (2017). These fields give a stable BL experiencing strong (approximately 10-15 m s$^{-1}$) N-S winds. A humidity inversion, coinciding with the BL temperature inversion, can be seen in the initial $Q_{vap}$ field (Fig. 1**a**):

previous studies (e.g. Curry et al., 1988; Solomon et al., 2011) have shown that such inversions are often observed in the Arctic, and may act as a source of moisture to BL clouds below. Surface sensible and latent heat fluxes were calculated by the model, using near-surface $\Theta$ and $Q_{vap}$ values, to represent an oceanic surface. The single-moment version of the Morrison et al. (2005) microphysics scheme was used, providing single-moment liquid (with a prescribed droplet number) and double-moment ice, snow, graupel, and rain.

In LES models, large-scale subsidence ($W_{sub}$) is often imposed as a tuning factor to maintain cloud top height. In such models, $W_{sub}$ is usually calculated from an imposed large-scale horizontal divergence. In practice, a constant divergence is assumed below the BL temperature inversion – with zero divergence above – producing a linear increase in $W_{sub}$ with height below the inversion, and a constant vertical wind above (Ovchinnikov et al., 2014; Solomon et al., 2015). Here, we calculate $W_{sub}$ using this method, increasing linearly with altitude up to 1500 m. At altitudes >1500 m, $W_{sub}=W_{sub}(1500$ m$)$

(representing zero divergence aloft).



**Table 1.** Simulation list.

| Test number | Run label | Horizontal divergence [s$^{-1}$] | Prescribed $N_{drop}$ [cm$^{-3}$] | $N_{ice}$ parameterisation | Surface forcing [Y/N]* |
|---|---|---|---|---|---|
| 1 | CNTRL | OFF | 100 | D10 | N |
| 1 | LOSUB | 2.5×10$^{-6}$ | 100 | D10 | N |
| 1 | HISUB | 5.0×10$^{-6}$ | 100 | D10 | N |
| 2 | CNTRL_Ndrop50 / 150 | OFF | 50 / 150 | D10 | N |
| 2 | LOSUB_Ndrop50 / 150 | 2.5×10$^{-6}$ | 50 / 150 | D10 | N |
| 2 | HISUB_Ndrop50 / 150 | 5.0×10$^{-6}$ | 50 / 150 | D10 | N |
| 3 | CNTRL_D10x0.5 / 2 | OFF | 100 | D10×0.5 / 2 | N |
| 3 | LOSUB_D10x0.5 / 2 | 2.5×10$^{-6}$ | 100 | D10×0.5 / 2 | N |
| 3 | HISUB_D10x0.5 / 2 | 5.0×10$^{-6}$ | 100 | D10×0.5 / 2 | N |
| 4 | CNTRL_SURFWARM | OFF | 100 | D10 | Y |
| 4 | LOSUB_SURFWARM | 2.5×10$^{-6}$ | 100 | D10 | Y |
| 4 | HISUB_SURFWARM | 5.0×10$^{-6}$ | 100 | D10 | Y |

* See text for further details.

In the literature, the imposed horizontal divergence in LES studies often ranges from $2.5\times10^{-6}\,s^{-1}$ (Solomon et al., 2015), through $3.75\times10^{-6}\,s^{-1}$ (Wang and Feingold, 2009a; Feingold et al., 2015; Yamaguchi and Feingold, 2015), to $5\times10^{-6}\,s^{-1}$ (Ovchinnikov et al., 2011). In this study, three different levels of imposed divergence – $0\,s^{-1}$, $2.5\times10^{-6}\,s^{-1}$, and $5\times10^{-6}\,s^{-1}$ – are used in four separate tests to investigate the role of large-scale subsidence in both stable and precipitation-favouring microphysical scenarios. Details of the tests conducted are listed in Table 1. Test 1 (Sect. 3.1) considers the effect of imposing different levels of subsidence on the microphysical properties of a stable mixed-phase Sc layer. In Sects 3.2 and 3.3, parameters relating to development of precipitation in the liquid or ice phase are varied to test the microphysical response under different levels of large-scale subsidence. For example, by decreasing droplet number concentrations ($N_{drop}$, Sec 3.2), we expect to enhance rain formation with little effect on snow (test 2). The control simulations apply no large-scale subsidence, a prescribed $N_{drop}$ of $100\,cm^{-3}$, and use the DeMott et al. 2010 (hereafter, D10) parameterisation for primary ice nucleation. As in Young et al. (2017), an approximation of the D10 parameterisation is used, where we assume an aerosol particle number concentration of $2.20\,cm^{-3}$ (for implementation in the parameterisation) throughout the domain. This approximation produces a relationship dependent solely on model temperature.

Test 4 investigates larger-scale BL interactions with a stable mixed-phase Sc layer. In CAOs, clouds move southwards off the sea ice and thus are subjected to a warming ocean surface. Model simulations in tests 1, 2, and 3 do not include any surface forcing: surface temperatures are allowed to vary through feedbacks with the BL above, yet they are not monotonically forced to become warmer. Such a forcing is applied in test 4 to investigate the combined influence of imposed large-scale subsidence and a warming surface. Surface temperatures are kept constant at 263.48 K until 5 h to allow adequate time for model spin up, after which they are forced to warm linearly, in hourly increments, to 265.66 K at approximately 11 h 20 min. This warming profile



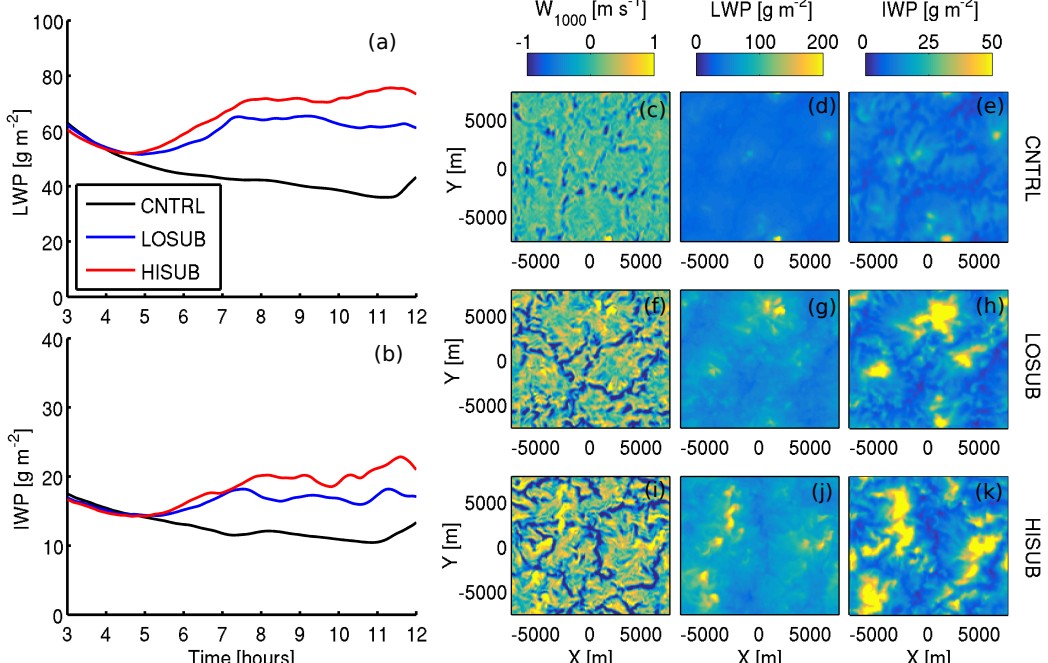

**Figure 2. (a, b):** Time series of the domain-averaged LWP and IWP from simulations imposing different magnitudes of large-scale subsidence. **Black:** Control cases, **blue:** low $W_{sub}$ (LOSUB), **red:** high $W_{sub}$ (HISUB). **(c–k):** Planar X-Y views of **(c,f,i)** vertical velocity at 1000 m ($W_{1000}$), **(d,g,j)** LWP, and **(e,h,k)** IWP. Planar views shown at 11 h.

was artificially constructed based on approximated ERA-Interim (ECMWF Reanalysis, Dee et al., 2011) 2 m temperature variations over the ocean in the Svalbard archipelago, close to the sea ice, during a cold air outbreak (23-Mar-2013, see Young et al., 2016, 2017, and Fig. S1 for further details).

We employ a horizontal resolution of 120 m over a domain of size 16 km×16 km. Vertical resolution for the majority of model simulations was 20 m up to 1500 m, decreasing to 50 m between 1500 m and 3000 m (domain lid) to reduce computational cost. A second domain structure was tested to check sensitivities to this set up: the high resolution region was extended to 2300 m (again, reducing to 50 m above this height). Whilst our results are largely unaffected by the introduction of more vertical levels (not shown, see Fig. S2), this modified domain structure was applied in Sect. 3.4 (test 4) due to increasing cloud height.



## 3  Results

### 3.1  Test 1: Stable stratocumulus

Firstly, the influence of large-scale subsidence on the evolution of a stable mixed-phase marine Sc is examined. Prescribed droplet number concentrations and parameterised primary ice nucleation were not altered. In all cases, the modelled clouds display the typical representation of an liquid-topped Arctic single-layer mixed-phase Sc, with heterogeneous ice number concentrations spread throughout the cloud below (not shown, Figs. S3–S5). Domain-averaged liquid- and ice-water paths (LWP, IWP) are shown in Fig. 2(**a,b**), where the first 3 h of each simulation is excluded due to model spin up. A stable Sc is modelled in the absence of $W_{sub}$ (CNTRL, Fig. 2**a**). Increasing $W_{sub}$ (LO- and HISUB) increases both the LWP and the IWP after approximately 5 h. These traces become more variable with time when subsidence is imposed, as is particularly visible in the IWP traces, suggesting increased dynamic activity in the modelled clouds. Longwave (LW) radiative cooling is instrumental in allowing this convection to develop in these clouds (see Supplement, Fig. S6).

Planar X-Y views of the vertical velocity at 1000 m ($W_{1000}$), LWP, and IWP fields (at 11 h), shown in Fig. 2(**c-k**), further illustrate the effect subsidence has on the spatial structure of the clouds. With increasing $W_{sub}$, numerous regions of high LWP/IWP develop, with heightened heterogeneity across the domain. Domain-wide variability in $W_{1000}$ also increases with imposed subsidence. Broad updraught regions surrounded by narrow downdraught rings become apparent. Localised regions of high LWP and IWP can be associated with strong updraughts at 1000 m, and lower IWPs mirror the shape of the downdraught rings around the updraught regions. This locality becomes clearer with increasing $W_{sub}$ (Fig. 2**i,k**).

Figure 3 illustrates the time series of total (resolved + sub-grid) turbulent kinetic energy (TKE, following Curry et al., 1988) and relative humidity (RH) in panels a–c, and vertical profiles of the snow + graupel tendencies, rain tendencies, total water mixing ratio ($Q_{tot}$), ice-liquid potential temperature ($\Theta_{il}$), time-averaged total vertical velocity variance ($w'^2$), water vapour flux ($w'Q_{vap}'$) and buoyancy flux ($w'\Theta'$) in panels d–i. $w'^2$ is used as an indicator for circulation strength, whilst the total (advected plus sub-grid) water vapour and buoyancy fluxes illustrate the mean dynamical motions in the BL. Here, a combined measure of sub-grid and advected fluxes is presented as these are of similar orders of magnitude and both make a non-negligible contribution to the flux budget (not shown, Fig. S7). In particular, the sub-grid $w'Q_{vap}'$ fluxes are dominant in-cloud and near the surface, due to the stability of these layers.

Strong snow sublimation is simulated at cloud top in all cases (Fig. 3**d**), with steady snow production in- and below-cloud. Increasing $W_{sub}$ has little effect on the snow tendencies. Non-zero snow rates reach the surface, suggesting heterogeneity in the snow field across the domain. In contrast, all of the modelled rain evaporates below cloud (Fig. 3**e**). Rain evaporation is strong at cloud top and base in all simulations; however, both the rain evaporation and production rates increase consistently with increasing $W_{sub}$ (listed in Table 2). Heightened rain evaporation below cloud coincides with increased below-cloud BL humidity (Fig. 3**b, c**) and increased $w'^2$ in the subsidence cases (Fig. 3**g**); however, the CNTRL simulation maintains a moist surface layer which could be acting as a source of moisture to the sub-cloud layer above it (<500 m, Fig. 3**a**). Additionally, the temperature and humidity inversion at the top of the BL acts to introduce a downward flux of heat and moisture into cloud top in all cases; however, the moisture flux in particular increases slightly with increased levels of $W_{sub}$.



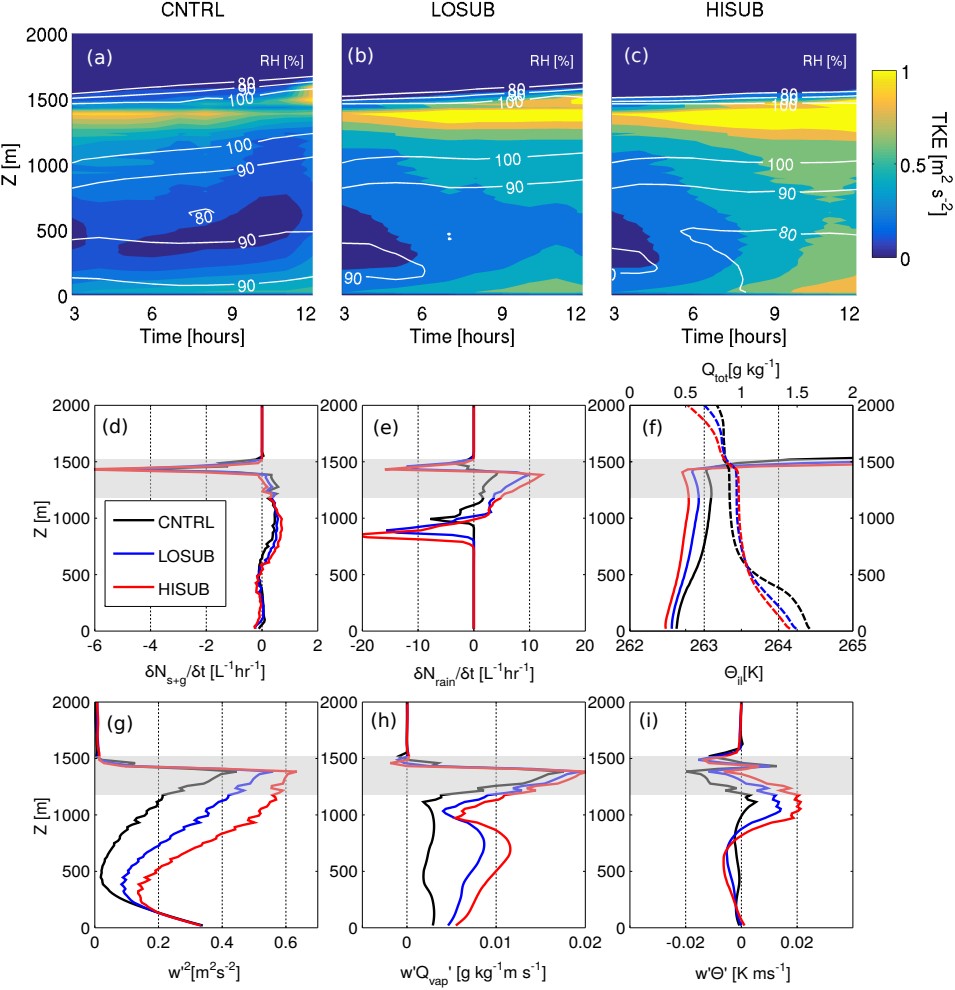

**Figure 3. (a–c):** Total turbulent kinetic energy (TKE, shading) and relative humidity (RH, white contours) time series for differing levels of subsidence. **(d–i):** Vertical profiles, at 9 h, of **(d):** solid precipitation (snow + graupel) tendency, **(e):** rain tendency ($\delta N_{rain}/\delta t$), **(f):** ice-liquid potential temperature ($\Theta_{il}$, solid) and total water mixing ratio ($Q_{tot}$, dashed) **(g):** vertical velocity variance ($w'^2$), **(h):** vertical flux of water vapour ($w'Q_{vap}'$) and **(i):** buoyancy flux ($w'\Theta'$). **(g–i):** $w'^2$, $w'Q_{vap}'$, and $w'\Theta'$ are total (sub-grid + advected) time-averaged quantities. Area in grey represents CNTRL cloudy regions.

Modelled ice-liquid potential temperatures ($\Theta_{il}$, following Tripoli and Cotton, 1981; Bryan and Fritsch, 2004) in the LO- and HISUB cases are colder than the CNTRL throughout the BL (Fig. 3f). In the region of strong rain evaporation (750 m–1200 m), below-cloud $\Theta_{il}$ tends towards an approximately neutral profile in all cases. All cases display a stable BL structure in the lower 1200 m of the BL, and an unstable structure within cloud. A minor inversion is modelled at approximately 500 m





**Table 2.** Key BL and cloud microphysical parameters affected by large-scale subsidence in test 1. Minimum $\delta N_{rain}/\delta t$ values correspond to below-cloud rain evaporation, whilst max $\delta N_{rain}/\delta t$ values relate to rain production within the cloud layer. $\Delta\Theta_{il}$ is calculated across the BL inversion and is listed to illustrate the inversion strength.

| Run label | Peak TKE[a,b] $[m^2 s^{-2}]$ | $\Delta\Theta_{il}$ [K] | Peak LWP[b] $[g\ m^{-2}]$ | Peak IWP[b] $[g\ m^{-2}]$ | Min / Max $\delta N_{rain}/\delta t$[c] $[L^{-1}\ hr^{-1}]$ |
|---|---|---|---|---|---|
| CNTRL | 1.0 | 7.52 | 62.9 | 17.5 | -7.8 / 4.3 |
| LOSUB | 1.3 | 7.74 | 65.4 | 18.2 | -15.7 / 9.8 |
| HISUB | 1.7 | 7.84 | 75.6 | 22.8 | -21.9 / 12.2 |

[a] At cloud top.

[b] Maximum values attained within 12 h simulation time.

[c] At 9 h (comparable with Fig. 3).

in the CNTRL case which is co-located with both a total water mixing ratio ($Q_{tot}$) inversion and the top of the moist surface layer in the CNTRL case.

Subsidence acts to make the temperature inversion stronger, as shown in Table 2, thus reducing entrainment into the cloud from above the BL. TKE increases throughout the BL with increasing subsidence (Fig. 3**b, c**) and peaks at cloud top in all cases, likely influenced by the high evaporation and sublimations rates of rain and snow at the BL-capping temperature inversion. In all simulations, TKE typically increases with altitude through the BL. When subsidence is imposed, these TKE profiles tend towards a coupled, well-mixed BL through the top-down and bottom-up propagation of TKE. This coupling is particularly clear in the HISUB case (Fig. 3**c**); however, the cloud top peak in TKE remains dominant throughout every case. Therefore, increasing $W_{sub}$ produces a more coupled, dynamic BL due to a heightened LWP, efficient LW radiative cooling, and increased rain evaporation below cloud.

### 3.2 Test 2: Droplet number concentration

The influence of large-scale subsidence on the formation of rain in a mixed-phase marine Sc is now considered. Prescribed droplet number concentrations were varied to a lower ($N_{drop} = 50\,cm^{-3}$) and higher ($N_{drop} = 150\,cm^{-3}$) threshold to affect rain formation: the modelled liquid mass is distributed amongst this concentration, such that a lower (higher) concentration will yield larger (smaller) cloud drops. Therefore, we expect the lower concentration of cloud droplets to allow for more efficient rain formation. Sandu and Stevens (2011) conducted a similar sensitivity study when studying Sc-to-cumulus transitions with an LES model and found that decreasing droplet number concentrations, and enhancing precipitation, significantly affected the transition efficiency. Figure 4 shows the domain-averaged LWP and IWP for test 2, along with vertical profiles of time-averaged total w'Θ', w'$Q_{vap}$', and w'$^2$ at 9 h. Simulations shown in variations of grey and black have no large-scale subsidence imposed. LOSUB cases are shown in variations of blue, whilst HISUB cases are shown in variations of red.

Considering the CNTRL cases, a decrease in $N_{drop}$ allows for a slightly greater LWP to be produced (Fig. 4**a**). As expected, droplets are able to grow to larger sizes and carry a greater mass as there are less sites available for condensation, producing a greater vertically-integrated liquid water mass (LWP). Modelled IWP is also greater after 8 h than the higher $N_{drop}$ cases





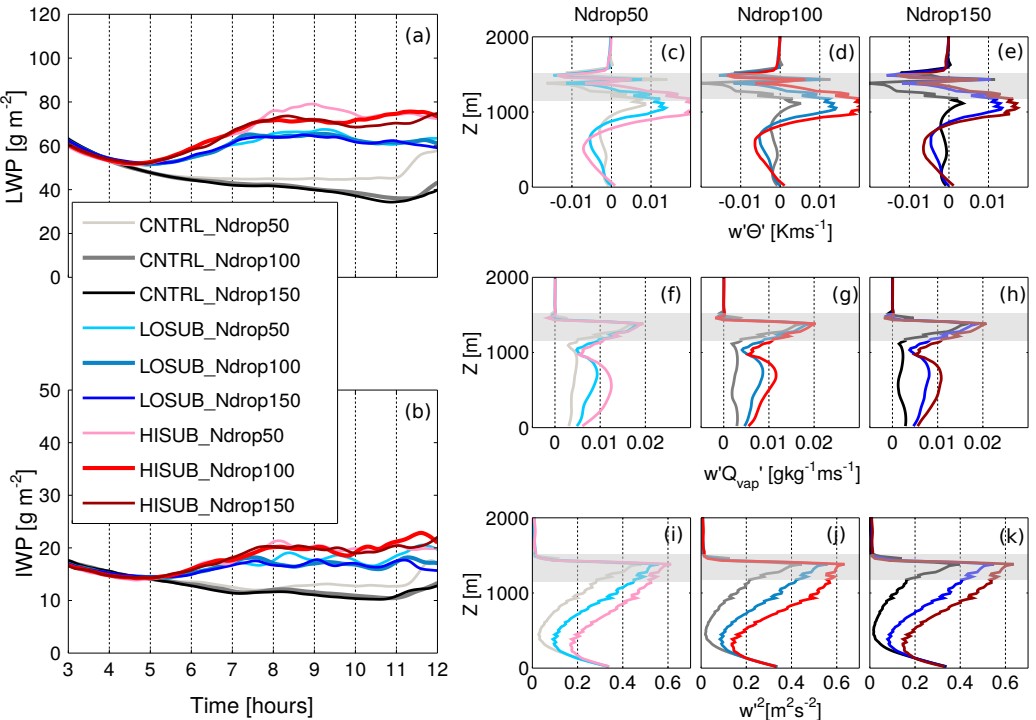

**Figure 4.** LWP **(a)** and IWP **(b)** time series for simulations with different $N_{drop}$ while varying the imposed large-scale subsidence. **Black:** Control cases, **blue:** low $W_{sub}$, **red:** high $W_{sub}$. **(c-e)**: buoyancy flux (w'$\Theta$'), **(f-h)**: water vapour flux (w'$Q_{vap}$'), **(i-k)**: vertical velocity variance (w'$^2$). Vertical profiles shown at 9 h.

(Fig. 4**b**). Increasing $N_{drop}$ has little effect on the LWP or IWP; the results of CNTRL_Ndrop100 and CNTRL_Ndrop150 are remarkably similar. Changing $N_{drop}$ has little effect on the depth of the cloud layer modelled in the CNTRL cases (shown by shading in Fig. 4**c-k**). Decreasing $N_{drop}$ dynamically influences the modelled cloud by producing a greater w'$\Theta$' within, and below, the cloud layer, suggesting a net upward flux of warm air (Fig. 4**c-e**). Below approximately 1000 m, all CNTRL

5   simulations display a cooling profile, coinciding with a positive upward flux of water vapour. As with test 1, the vapour profiles of all cases are influenced by the downward flux of warm, moist air from above the inversion. Turbulent activity is greater throughout the BL when less droplets are prescribed (CNTRL_Ndrop50, Fig. 4**i**), whereas the w'$^2$ profile is dominated by sharp peaks at the surface and at the top of the BL when more droplets are modelled (CNTRL_Ndrop150, Fig. 4**k**).

    As with test 1, greater LWPs and IWPs are modelled with increasing $W_{sub}$. The microphysical changes (varying $N_{drop}$)

10   affect the modelled LWP and IWP less than varying $W_{sub}$ (Fig. 4**a,b**). The LO- and HISUB simulations are again more dynamic throughout the BL than the CNTRLs (Fig. 4**i–k**). w'$^2$ increases with height throughout the BL with increasing $W_{sub}$ – excluding a minimum modelled at approximately 500 m in each case – and a greater w'$Q_{vap}$' is modelled below cloud. Peak in-cloud water vapour fluxes are largely unchanged with $W_{sub}$ (Fig. 4**c–e**). A more substantial difference can be identified in the



buoyancy profiles: w'Θ' is much greater within cloud, and more negative below approximately 750 m, than the corresponding CNTRL simulations. This positive w'Θ' is likely due to a downward flux of cold air, suggesting these cases are dominated by strong downdraughts; downdraughts which likely facilitate precipitation production.

Figure 5 shows the production and sublimation/evaporation rates of snow+graupel and rain relative to the CNTRL in panels

A and B respectively. Absolute domain-averaged number concentrations from each subsidence simulation are overlaid as contours. Varying $N_{drop}$ has only a minor effect on the time evolution of $N_{s+g}$. With increasing subsidence, $N_{s+g}$ decreases slightly in, and increases below, cloud. Similarly, snow production rates (relative to the CNTRL) are greater in and directly below the cloud layer with increasing $W_{sub}$, whilst snow sublimation rates at cloud top also increase. Non-zero snow concentrations reach the surface in all simulations (Fig. 5**A**).

Figure 5**B** shows a contrasting trend for rain production and evaporation. As with test 1, strong rain evaporation at cloud top and base is offset by high production rates within the cloud layer. Decreasing $N_{drop}$ strongly affects $N_{rain}$ as expected; for example, the rain number concentration increases by approximately $10 \, L^{-1}$ between the HISUB_Ndrop100 and HISUB_Ndrop50 cases. For the LOSUB comparison, $N_{rain}$ increases by approximately $6 \, L^{-1}$ in cloud. Increasing $W_{sub}$ therefore increases the number concentration of rain particles produced by decreasing $N_{drop}$ in the modelled cloud. The moist, cool sub-cloud layer

and increased w'$^2$ at cloud base shown in Fig. 4 coincide with the top of these regions of increased rain evaporation below cloud in all subsidence cases.

Increasing $N_{drop}$ has a smaller effect on $N_{rain}$ as expected by the thermodynamic indirect effect; with more droplets available, droplet size decreases due to less competition for water vapour. $N_{rain}$ decreases in Ndrop150 with respect to the Ndrop100 or Ndrop50 cases, and the in-cloud production and below-cloud evaporation rates are smaller. Despite this, increasing $W_{sub}$

still increases the production/evaporation rates with respect to the CNTRL_Ndrop150 case.

No rain reaches the surface in any of these simulations; all of the $N_{rain}$ produced evaporates directly below cloud. This evaporation effect increases with increasing subsidence; for example, the HISUB_Ndrop50 case evaporates at $-68 \, L^{-1} \, hr^{-1}$ below cloud at 9 h. From these simulations, we suggest that the level of imposed large-scale subsidence can significantly affect the liquid phase in clean mixed-phase Sc, as $W_{sub}$ positively forces the rain production/evaporation rates modelled in these

precipitation-favouring microphysical scenarios.

### 3.3 Test 3: Ice number concentration

The influence of $W_{sub}$ on a mixed-phase marine Sc when changing ice number concentrations is now considered. Heterogeneous primary ice formation is represented using the D10 parameterisation with aerosol number concentrations calculated during the study by Young et al. (2017). Several previous studies (Harrington et al., 1999; Harrington and Olsson, 2001; Prenni

et al., 2007; Morrison et al., 2012; de Boer et al., 2011; Young et al., 2017) have shown that the lifetime of springtime single-layer mixed-phase clouds at high latitudes is strongly dependent on $N_{ice}$. Here, a lower ($N_{ice}$ = D10×0.5) and higher ($N_{ice}$ = D10×2) threshold are implemented to change the number concentration of modelled ice, and snow, particles. We expect increasing $N_{ice}$ to increase $N_{s+g}$, as the cold temperatures will allow for efficient vapour growth of ice particles into snow at the expense of the cloud liquid phase by the WBF mechanism. However, decreasing $N_{ice}$ should sustain the liquid phase



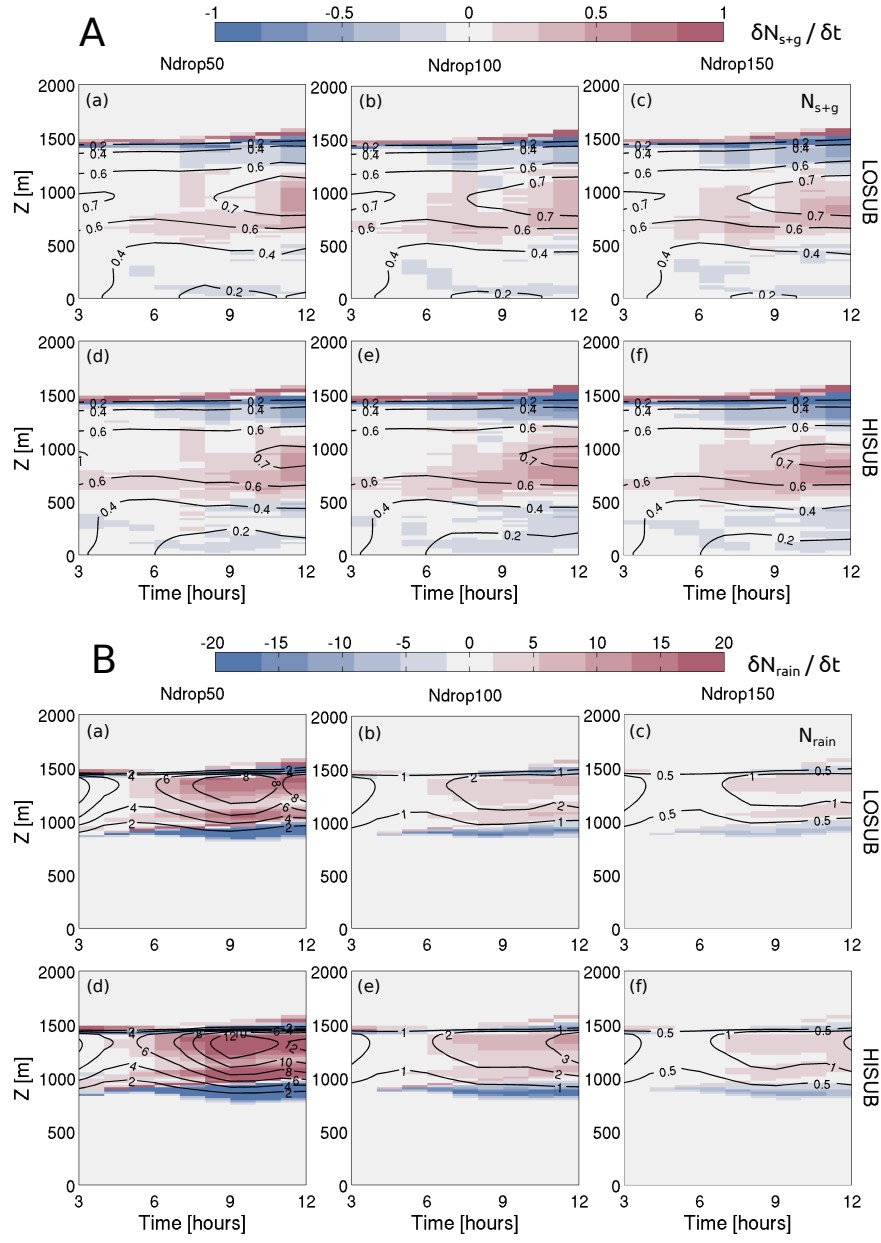

**Figure 5. A:** Change in $\delta N_{s+g}/\delta t$ [$L^{-1}$ $hr^{-1}$] (shading) between subsidence cases and the corresponding CNTRL simulation for test 2. Red corresponds to increased production, whilst blue shows increased sublimation than the associated CNTRL. $N_{s+g}$ [$L^{-1}$] is shown as contours. **B:** As panel A, instead the change in $\delta N_{rain}/\delta t$ [$L^{-1}$ $hr^{-1}$] is shown with $N_{rain}$ [$L^{-1}$] as contours. **(a-c):** LOSUB, **(d-f):** HISUB.

against the WBF mechanism, likely affecting rain formation. Therefore, this test has the potential to affect both rain and snow in the modelled clouds.





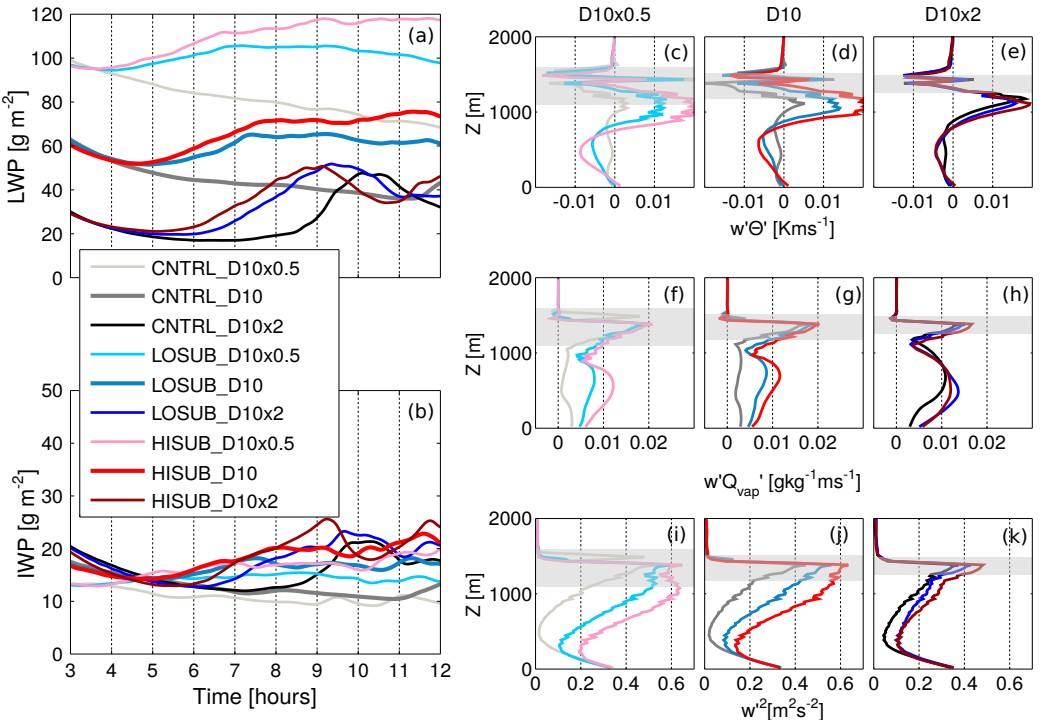

**Figure 6.** As Fig. 4, but with changing ice number concentrations.

Fig. 6 illustrates the domain-averaged LWP and IWP for test 3. The CNTRL cloud layer – as shown by the shaded area in Fig. 6**(c-k)** – becomes shallower with increasing $N_{ice}$. When no subsidence is imposed (CNTRL), decreasing $N_{ice}$ increases the LWP as expected through the influence of the WBF mechanism, whereas increasing $N_{ice}$ has the opposite effect (black/grey lines, Fig. 6**a**). However, in CNTRL_D10×2, both the LWP and IWP increase sharply after 9 h (Fig. 6**a,b**). This LWP peak

occurs more quickly with increasing $W_{sub}$ (as shown by the blue and brown traces in Fig. 6**a**) and, although the shape of the peak changes, this trend can also be seen in the IWP traces.

As with tests 1 and 2, the modelled LWP increases with increasing $W_{sub}$. Trios can be easily identified in Fig. 6**(a)**, where increasing the subsidence affects the LWP more so than altering $N_{ice}$. Imposing subsidence produces LWPs which are stable, or even increase, with time. Furthermore, increasing the subsidence whilst altering $N_{ice}$ marginally increases the modelled IWP

(Fig. 6**b**); however, the traces are similar between the varying $N_{ice}$ and subsidence scenarios tested.

Coinciding with these larger LWPs, below-cloud $w'^2$ increases with increasing subsidence and decreasing $N_{ice}$. Additionally, the extremes in the modelled $w'\Theta'$ profile are more exaggerated in the LO- and HISUB cases. The exception to this trend is the high ice (D10×2) simulations, as subsidence does not affect this microphysical scenario as strongly as in the D10×0.5 cases. The modelled clouds are less dynamic with a larger $N_{ice}$; $w'^2$ becomes dominated by cloud top and surface peaks in the

CNTRL_D10x2 case, similar to CNTRL_Ndrop150 in test 1. These comparisons suggest that $W_{sub}$ has a strong dynamical





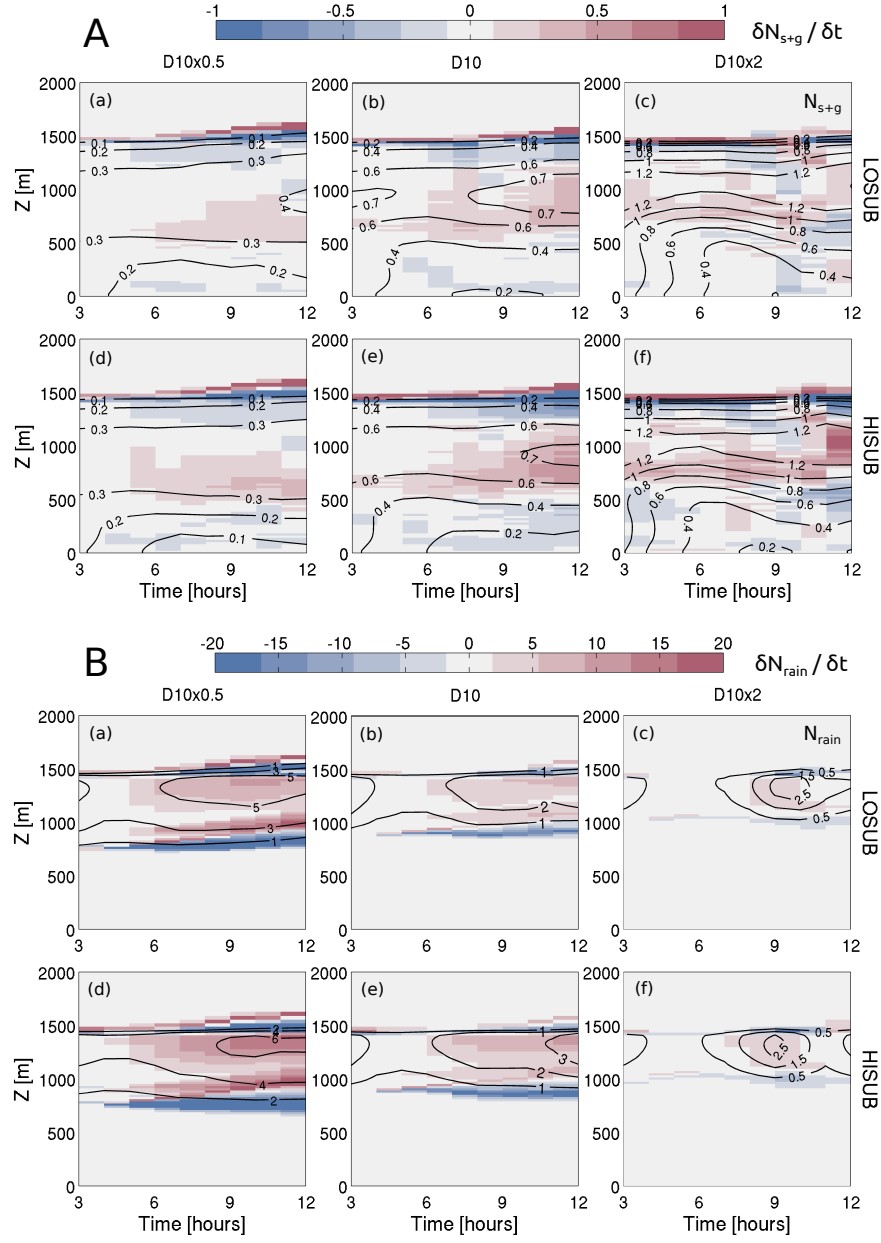

**Figure 7.** As Fig. 5, instead the ice number concentration is varied (test 3).

effect on liquid-dominated mixed-phase clouds, but have little influence in those with more ice. With less ice (D10×0.5), the $w'^2$ peak at cloud top grows larger and deeper in comparison to the D10×2 simulations, indicating that a more dynamic cloud-topped BL is modelled when less ice is present.



$N_{s+g}$ increases with increasing $N_{ice}$ as expected, and decreases only slightly between the LO- and HISUB cases (Fig. 7**A**). Whilst the absolute number concentrations marginally decrease, the in-cloud snow production rates – relative to the CNTRL simulations – increase with increasing subsidence, with up to $1\,\mathrm{L}^{-1}\,\mathrm{hr}^{-1}$ at 12 h (1000 m) modelled in the HISUB_D10x2 case (Fig. 7**A(f)**). Additionally, snow sublimation rates at cloud top and below cloud increase with increased $W_{sub}$, similar to test 2.

As with tests 1 and 2, non-zero $N_{s+g}$ reaches the surface (Fig. 7**A**), whilst all rain evaporates in the sub-cloud layer (Fig. 7**B**). LOSUB_D10x2 actually produces a greater $N_{rain}$ at approximately 9 h than LOSUB_D10, whereas the HISUB cases produce less rain with increased $N_{ice}$ as expected. This LOSUB_D10x2 artefact likely occurs due to the lower below-cloud rain evaporation rates than LOSUB_D10, and similar in-cloud production rates, allowing more rain to remain in the less-dynamic cloud (not shown, Fig. S8).

In test 2, increasing $W_{sub}$ dynamically stimulated a scenario with inefficient precipitation production (HISUB_Ndrop150). Interestingly, $W_{sub}$ does not have this same effect here when increasing $N_{ice}$; increasing $W_{sub}$ does not efficiently generate convection in LO- and HISUB_D10x2, and the modelled $\delta N_{rain}/\delta t$ rates vary little in comparison to the D10×0.5 and D10 cases. The consistency between the LO- and HISUB_D10x2 cases is also demonstrated in the small difference between the w'Θ', w'$Q_{vap}$', and w'$^2$ profiles at 9 h (Fig. 6**e, h, k**). Increasing $N_{ice}$ appears to suppress the formation of $N_{rain}$ by acting as a sink for water vapour and causing droplet evaporation (via the WBF mechanism), thus suppressing rain-driven convection.

Greater LWPs are produced with less ice, as more liquid mass is able to form in the vicinity of ice crystals, producing larger droplets (for a fixed $N_{drop}$). With larger droplets, more efficient rain production can take place, as shown in Fig. 7**B**. Rain production and evaporation rates increase strongly with increasing $W_{sub}$ in the D10×0.5 case. Below-cloud rain evaporation rates increase with decreased $N_{ice}$ – $-40\,\mathrm{L}^{-1}\,\mathrm{hr}^{-1}$ at 12 h (700 m) in the HISUB_D10x0.5 case – as do the in-cloud production rates and $N_{rain}$. With less ice available, more liquid droplets may form and grow against a weaker WBF mechanism. Cloud top radiative cooling becomes more efficient due to a heightened liquid fraction (Fig. 6**a**), vigorous turbulence (Fig. 6**i**), and rain formation (Fig. 7**B(d)**). Consequently, cloud top height increases in D10×0.5, while this ascent is suppressed in D10×2. w'$^2$ is greatest with the D10×0.5 simulations (Fig. 6**i**) due to the dynamical activity produced by the heightened rain evaporation at cloud base (Fig. 7**B**), and this effect is strengthened with increasing $W_{sub}$. These conclusions support the findings of test 2, as it is the liquid-dominated clouds which are vulnerable to dynamic stimulation by imposing large-scale subsidence. Clouds with greater ice number concentrations suppress the liquid phase; therefore, the ice number concentration has a key role in mediating the strength of turbulent overturning generated in the mixed-phase clouds.

### 3.4 Test 4: Surface warming

As described in Sect. 2, our previous tests consider scenarios that would elicit a microphysical response whilst keeping the surface boundary conditions approximately constant. These scenarios give an indication of how subsidence can affect precipitation formation in mixed-phase clouds that remain at approximately the same latitude. Tests 1–3 are idealised and are not representative of the environmental forcings encountered when these clouds move southwards: observations show a sharp near-surface air temperature gradient in CAO flows transitioning from the cold sea ice to the warm ocean. To address this, we further consider the combined dynamical impact of large-scale subsidence and a warming surface on both the BL and cloud





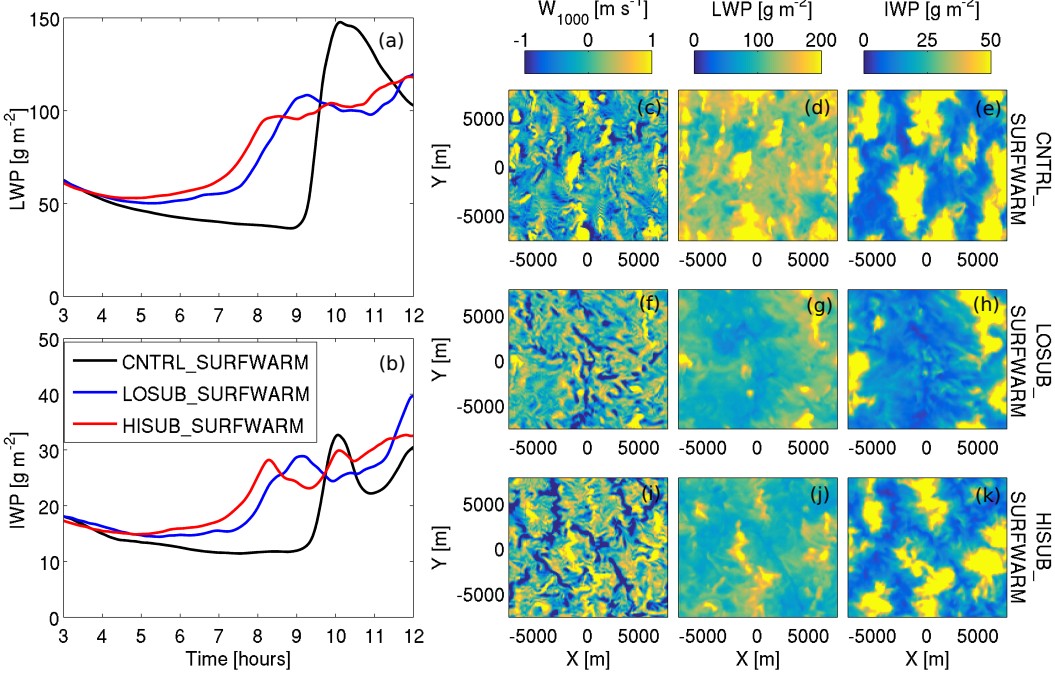

**Figure 8.** As Fig. 2, with the addition of a warming surface (test 4). Planar views shown at 10 h to capture the bulk cloud structure coinciding with the CNTRL_SURFWARM peak in LWP and IWP shown in panels **a** and **b**.

microphysical structure. Whilst our domain size is not appropriate to resolve the explicit transition from closed to open cellular convection far downstream in a CAO, we will show how large-scale subsidence influences the microphysical stability of a stable mixed-phase marine Sc over a warming surface, upstream from this strong cellular convection.

The simulated clouds become more convective with time under the destabilising conditions of a warming surface (Fig. 8).
This process is gradual when subsidence is imposed, as shown by the approximately monotonic increase in LWP and IWP with time; however, a sharp increase in both LWP and IWP is modelled in the CNTRL_SURFWARM case at 10 h. The planar views of Fig. 8(**c–e**) show that, at this time, the CNTRL_ SURFWARM cloud contains numerous regions of very high LWP ($>200\,\mathrm{g\,m^{-2}}$) and IWP ($>50\,\mathrm{g\,m^{-2}}$) co-located with strong updraughts at 1000 m. The differences between the LO- and HISUB cases are not as prominent as without surface forcing (test 1, Fig. 2); however, in agreement with the test 1 simulations,
subsidence again causes an increase in both LWP and IWP with time, and produces greater values than CNTRL_SURFWARM until 9 h when the control undergoes a significant convective transformation.

Figure 9 mirrors the format of Fig. 3 to allow a direct comparison of the influence of a warming surface; however, TKE data are shown over a greater colour range in Fig. 9(**a–c**). In contrast to Fig. 3, cloud top and surface sources of TKE couple in all cases. The CNTRL case couples rapidly at approximately 10 h (Fig. 9), coincident with the peak in LWP and IWP shown in
Fig. 8(**a,b**). Cloud top and surface TKE sources then appear to decouple at approximately 11 h. Cloud top and surface sources




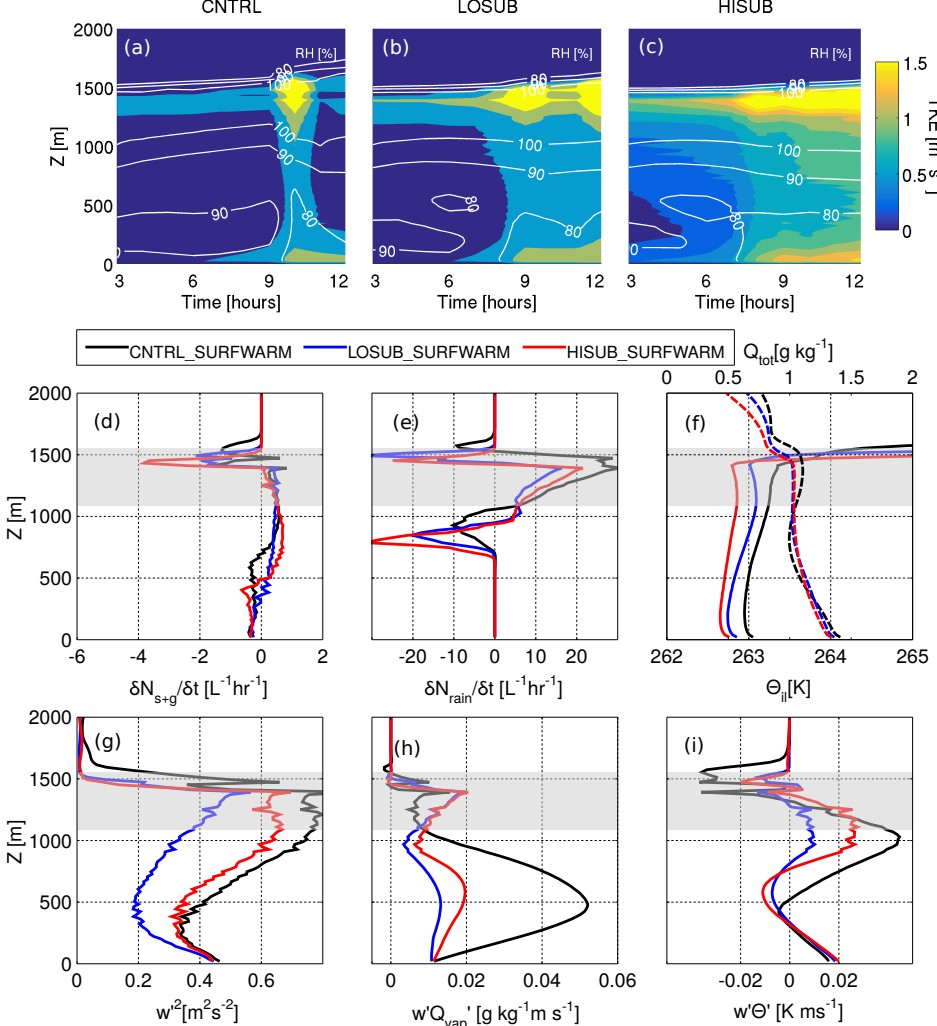

**Figure 9.** As Fig. 3, with the addition of a warming surface (test 4). Note the different colour scale in panels **a–c**, and extended x-range over which data is shown in panel **e**, with comparison to Fig. 3. Vertical profiles (panels **d–i**) are shown at 10 h.

of TKE again separately dominate the LO- and HISUB_SURFWARM profiles from approximately 7 h onwards; however, the surface contributions are stronger than modelled in test 1. LOSUB_SURFWARM displays a similar coupling at 10 h to CNTRL_SURFWARM, yet it remains coupled afterwards and undergoes a second TKE burst between 11 h and 12 h. TKE evolves similarly in HISUB_SURFWARM as in HISUB (test 1, Fig. 3c), with the propagation of top-down and bottom-up

5 TKE gradually increasing with time to couple the separated cloud and surface sources. The warming surface acts to produce an instability in each of the $\Theta_{il}$ profiles at the surface (Fig. 9f).


**Table 3.** Key BL and cloud microphysical parameters affected by large-scale subsidence in test 6. As Table 2, minimum $\delta N_{rain}/\delta t$ values correspond to below-cloud rain evaporation, whilst max $\delta N_{rain}/\delta t$ values relate to rain production within the cloud layer.

| Run label | Peak TKE[a,b] $[m^2 s^{-2}]$ | $\Delta\Theta_{il}$ [K] | Peak LWP[b] $[g\,m^{-2}]$ | Peak IWP[b] $[g\,m^{-2}]$ | Min / Max $\delta N_{rain}/\delta t$[c] $[L^{-1}\,hr^{-1}]$ |
|---|---|---|---|---|---|
| CNTRL | 2.8 | 7.63 | 147.7 | 32.7 | -10.2 / 30.6 |
| LOSUB | 3.9 | 8.06 | 119.8 | 39.6 | -20.2 / 16.3 |
| HISUB | 2.3 | 8.37 | 118.3 | 32.7 | -31.6 / 21.5 |

[a] At cloud top.

[b] Maximum values attained within 12 h simulation time.

[c] At 10 h (comparable with Fig. 9).

Increased snow sublimation is modelled towards the surface in the surface warming cases (Fig. 9**d**), especially when subsidence is imposed. Cloud top height increases steadily in CNTRL_SURFWARM (Fig. 9**a**), whilst this ascent is strongly suppressed in HISUB_SURFWARM (Fig. 9**c**) and marginally suppressed in LOSUB_SURFWARM (Fig. 9**b**). Negative w'$\Theta$' fluxes at cloud top again suggest entrainment of warm air into the cloud layer from above the BL in each case; however, this

flux is stronger in CNTRL_SURFWARM than in the subsidence cases, indicating that greater entrainment rates are accompanying the cloud top ascent. Below-cloud and surface w'$Q_{vap}$' and w'$\Theta$' fluxes are stronger than in test 1, likely due to the increased turbulent overturning generated by the warming surface.

Qualitatively, the trends identified in test 1 remain true with a warming surface: below-cloud rain evaporation (Fig. 9**e**), BL TKE (Fig. 9**b,c**), $w'^2$ (Fig. 9**g**), and inversion strength (Table 3) are enhanced with increasing $W_{sub}$. However, the linearity

between w'$Q_{vap}$' and w'$\Theta$' with increasing $W_{sub}$ shown in test 1 does not hold true in this scenario: namely, significantly larger values ($0.052\,g\,kg^{-1}\,m\,s^{-1}$ and $0.045\,K\,m\,s^{-1}$, respectively) are modelled below cloud in the CNTRL_SURFWARM simulation than in the subsidence cases, coinciding with the rapid BL coupling shown in Fig. 9**(a)**. Convective activity increases at this time, with $w'^2$ increasing up to $0.90\,m^2\,s^{-2}$ in cloud alongside a peak (cloud top) TKE of $2.8\,m^{-2}$ (Table 3). Rain production is particularly strong in CNTRL_SURFWARM at this time (Table 3); however, below-cloud rain evaporation is still

weaker than in the LO- and HISUB_SURFWARM simulations. Similar to test 1, rain evaporative cooling below cloud in LO- and HISUB_SURFWARM again acts to decouple the surface and in-cloud heat sources from each other (Fig. 9**i**); however, the addition of a surface heat source causes the w'$\Theta$' profiles to swing through greater extremes; for example, from $0.021\,K\,m\,s^{-1}$ through $-0.011\,K\,m\,s^{-1}$ to $0.028\,K\,m\,s^{-1}$ in the HISUB_SURFWARM case.

The CNTRL_SURFWARM simulation experiences a sharp burst of TKE at cloud top 10 h, at which point the bottom-up

propagation from the surface appears to fully couple the BL. Within approximately 1.5 h, the TKE sources decouple again. Z-X slices of several microphysical variables are shown in Fig. 10 to sample the cloud structure at 10 h. In the bottom panel, below-cloud cumuli form which either couple to the Sc layer (white ellipses) or remain separate (red ellipses). These cumuli structures are clearly visible in the $Q_{liq}$ contour field (top panel, Fig. 10). Cumuli are identified by adjacent updraught/downdraught regions with 100 % RH (or close to 100 %). Cumuli can be seen from 8 h onwards, and become more frequent with time.

Two spatially-close cumuli form at approximately -7000 m and -3500 m which mark the boundaries of a detraining layer of





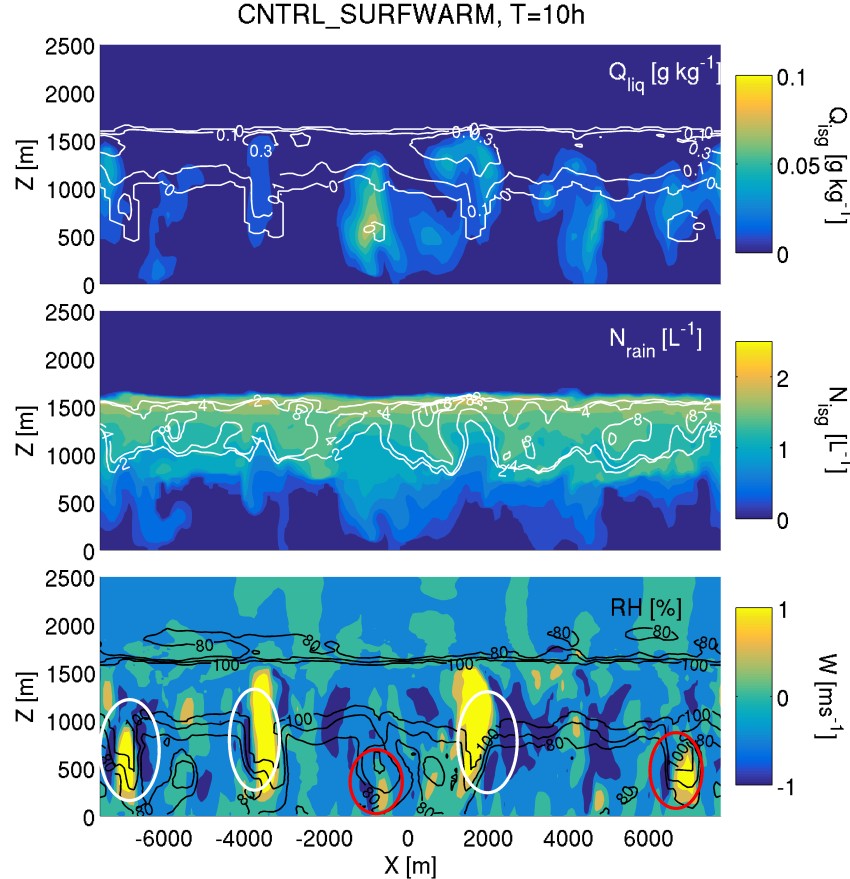

**Figure 10.** Z-X slices for the CNTRL_SURFWARM case at 10 h. **Top row:** total ice mass mixing ratio ($Q_{isg}$, shading) and liquid water mass mixing ratio ($Q_{liq}$, contours). **Middle row:** total ice number concentration ($N_{isg}$, shading) and rain number concentration ($N_{rain}$, contours). **Bottom row:** vertical velocity (W, shading) and relative humidity (RH, contours). Identified detached below-cloud cumuli are highlighted by red ellipses, and cumuli merged with the Sc are indicated by white ellipses.

moisture above cloud top. Additionally, a similar, completely detached moist layer can be seen above cloud top coinciding with the 6000 m cumulus.

Total ice number concentrations (ice + snow + graupel, $N_{isg}$, Fig. 10**b**) are largely unaffected by the dynamical stimulation of the cloud by the warming surface whereas, as previously found when imposing subsidence (test 1, Sect. 3.1), the liquid phase (both $Q_{liq}$ and $N_{rain}$) is positively influenced. In particular, $N_{rain}$ in CNTRL_SURFWARM is much more comparable with the corresponding domain-averaged values of the LO- and HISUB_Ndrop50 simulations in test 2 (Fig. 5**B**) – the efficient liquid precipitation cases – than any of the previous control simulations.

Similar to without surface forcing (Fig. S5), strong downdraughts form at lower altitudes in the Sc layer in HISUB_SURFWARM, likely forced by heightened evaporative cooling below cloud (Fig. 11). HISUB_SURFWARM has much larger updraught and



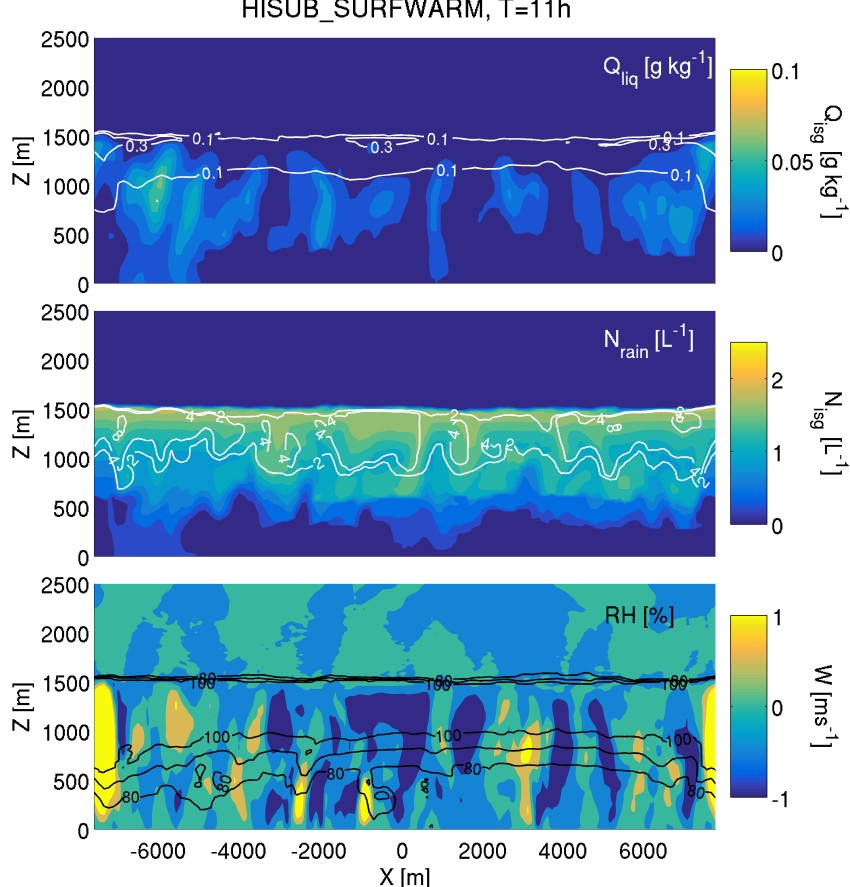

**Figure 11.** Z-X slices for the HISUB_SURFWARM case at 12 h. Panels are arranged similarly to Fig. 10.

downdraught regions than CNTRL_SURFWARM: from approximately 11 h onwards, these often extend to almost the full height of the BL. No distinct sub-cloud cumuli are identified in HISUB_SURFWARM (Fig. 11), whereas these are common in CNTRL_SURFWARM (Fig. 10): the addition of subsidence acts to suppress their formation and allow a more homogeneous Sc layer to be maintained in a BL undergoing top-down and bottom-up coupling of TKE. In test 1, this coupling was primarily

5  driven by the cloud TKE sources, whereas the extra input from the warming surface here leads to a more coupled TKE profile than without. The coupling process is more gradual in HISUB_SURFWARM than the CNTRL or LOSUB counterparts, suggesting that subsidence plays a role in whether or not this rapid TKE coupling and cloud top ascent can take place.



## 4 Discussion

### 4.1 Effect of subsidence on bulk cloud properties

Imposing large-scale subsidence in simulations of marine sub-Arctic mixed-phase Sc increases the LWP and IWP of the modelled clouds through increased convective activity throughout the domain (Fig. 2). $W_{sub}$ does not affect the cloud extent (Fig. 5, 7); only $N_{ice}$ notably affects the modelled cloud depth (Fig. 6). Dynamical stimulation by subsidence – which would sustain a mixed-phase Sc for longer against the WBF mechanism – may therefore have been previously missed in observations and modelling studies. Increasing $W_{sub}$ has a greater effect on the liquid phase than the ice phase (Figs. 2, 4, 6); however, increasing subsidence causes the development of heterogeneity in the LWP and IWP fields, leading to instabilities in the modelled clouds. In particular, the radiative properties of the clouds would be affected by the heterogeneous spread in LWP, where regions of high LWP would be more reflective to incoming shortwave radiation (Schröter et al., 2005) and more efficiently cooled via longwave radiative cooling. Localised regions of high IWP are typically co-located with updraughts in our simulations, likely due to the method of parameterising ice nucleation in our model. Namely, additional nucleation mechanisms (e.g. contact, immersion) are not represented such that we have a predictable source of ice number concentrations (similar to Young et al., 2017). These mechanisms would likely influence our results if they were explicitly resolved in our model; for example, we would expect contact nucleation in downdraughts, through interaction with interstitial aerosol particles.

Subsidence strongly influences the LWP; however, increasing levels of subsidence also marginally increase the domain-averaged IWP (Fig. 2**b**). Figure 6(**b**) shows that the peak IWP attained by CNTRL_D10x2 is also achieved in the HISUB_D10 case, suggesting that increasing $W_{sub}$ can have a similar effect on the bulk ice properties of the cloud as increasing $N_{ice}$. However, a much larger LWP is also modelled when subsidence is imposed, creating a microphysical structure that may be more robust against the WBF mechanism. This may allow mixed-phase conditions to be sustained for longer against a higher $N_{ice}$; a problem that is often faced when modelling Arctic mixed-phase Sc (Harrington and Olsson, 2001; Prenni et al., 2007; Morrison et al., 2012; de Boer et al., 2011, 2014; Young et al., 2017).

### 4.2 Effect of subsidence on microphysics and precipitation

In the chosen microphysical scenarios that may affect precipitation development in marine mixed-phase Sc, large-scale subsidence enhances rain evaporation at cloud top and base. Increased subsidence leads to larger rain production rates and a greater $N_{rain}$ within cloud, and this effect is particularly clear when lowering $N_{drop}$ (Ndrop50, Fig. 5) or lowering $N_{ice}$ (D10×0.5, Fig. 7) in tests 2 and 3 respectively. In these cases, the increase in $N_{rain}$ due to subsidence is less than can be attributed to the imposed microphysical changes; for example, an increase of approximately $6\,\mathrm{L^{-1}}$ is modelled in the Ndrop50 scenario due to increasing $W_{sub}$, whilst an increase of approximately $9\,\mathrm{L^{-1}}$ is achieved by lowering $N_{drop}$ from $100\,\mathrm{cm^{-3}}$ to $50\,\mathrm{cm^{-3}}$ (Fig. 5**B**).

From test 2, we can conclude that large-scale subsidence amplifies the modelled turbulence in scenarios allowing for efficient precipitation formation (Ndrop50, Fig 4**i**). In fact, $W_{sub}$ also acts to promote turbulence (Fig. 4**k**) and rain formation (Fig. 5**B**) in a microphysical scenario that produces little rain in its absence (Ndrop150). Conversely, increasing $N_{ice}$ in test 3 does





not have the same effect, and increasing $W_{sub}$ does little to promote turbulence in this scenario. Increased snow number concentrations and production/sublimation rates do not have the same dynamical effect on these clouds as similar changes in the rain category, and the combined cooling from rain evaporation at cloud base and radiative cooling at cloud top causes the efficient development of convection in these liquid-dominated clouds. These findings indicate that subsidence has the potential

to positively force the liquid phase of these clouds whilst having little effect on the ice phase. Young et al. (2016) presented observations of cloud microphysics over the transition from sea ice to ocean and found that the ice phase changed little under the dynamical evolution of the BL, while the liquid water content increased four-fold. Our findings therefore suggest that mixed-phase clouds with low number concentrations of primary ice, such as those commonly observed in the springtime Arctic, are vulnerable to dynamical changes induced by subsiding air from above or a warming surface from below.

Increasing $N_{ice}$ (test 3) produces more $N_{s+g}$ as expected, and this increase does not have the same dynamical effect as decreasing $N_{drop}$ (and producing more rain). However, whilst the ice categories do little to stimulate convection, they are responsible for suppressing rain formation; for example, a higher $N_{ice}$ (and thus, $N_{s+g}$) suppresses the strong rain production/evaporation processes modelled at a lower $N_{ice}$ (Fig. 7). With weakened rain formation and evaporation (Fig. 7**B**), less vigorous overturning is modelled in D10×2. Increased snow number concentrations, and production and sublimation rates, do

not have the same dynamical impact on these clouds as the production/evaporation of rain. Whilst the liquid phase drives the development of dynamical overturning, the ice phase has a strong influence – through the WBF mechanism – on whether this convective activity can actually develop.

All modelled rain does not reach the surface; in all of our simulations, strong rain evaporation occurs below cloud. These findings are in contrast to cloud-resolving model simulations of warm convective clouds by Feingold et al. (2015). Simulations

shown in Fig. 2 display a similar heterogeneity in W and LWP as the warm non-drizzling Sc case modelled by Savic-Jovcic and Stevens (2008). All rain produced by our simulations evaporates below cloud; therefore, they could be termed "non-drizzling". However, it is important to note that precipitation as snow is modelled in every case shown, and this snow always reaches the surface. Observational studies of Arctic marine mixed-phase Sc (Young et al., 2016) and North Atlantic CAOs (Abel et al., 2017) have previously reported precipitation as snow below cloud with little-to-no rain measured, indicating that our idealised

modelling study is in broad agreement with measurements in this region.

The LO- and HISUB_SURFWARM cases, like their test 1 counterparts, continue to produce heightened rain evaporation below cloud, introducing a negative w'Θ' flux to the middle of the BL. Weaker below-cloud rain evaporation occurs in CNTRL_SURFWARM, and the upward propagation of heat and moisture from the surface causes distinct cumuli to form below cloud and join with cloud base. These cumuli dynamically stimulate the cloud from below (Fig. 10) and have a similar

effect on the cloud as the introduction of subsidence in tests 1–3; for example, the warming surface allows a greater $N_{rain}$ to form in cloud (Fig. 10). In the absence of strong subsidence, namely in CNTRL and LOSUB_SURFWARM, the warming surface acts to push cloud top higher, and increase the LWP, through the formation of these below-cloud cumuli. This may suggest that, in regions of low subsidence, cloud top height may be forced upwards by a warming surface, causing strong heterogeneities to form in the spatial distribution of the LWP (Fig. 8**d**), leaving the cloud layer vulnerable to the formation of

strong convective cells with motion southwards.





### 4.3 Effect of subsidence on the BL and dynamics

Convective activity increases in the modelled clouds with $W_{sub}$ through increased BL TKE and below-cloud $w'^2$ in test 1–3. Solar heating acts to marginally offset the formation of defined closed-cellular structure; however, the cloud-driven convection is strongly dependent on cloud top LW radiative cooling (see Supplement, Fig. S6). Additionally, rain formation rates, number concentrations, and the domain-averaged LWP increase with increasing $W_{sub}$. This finding mirrors the conclusions of Hill et al. (2014), where the authors found that increasing the resolved TKE and/or temperature positively influences the liquid phase in ice saturated conditions, as these contribute towards sustaining water saturation.

With a larger LWP, stronger cloud top radiative cooling is expected, promoting a greater cloud top height (Wang and Feingold, 2009a). Subsidence acts to restrict cloud top ascent by reinforcing the BL temperature inversion (Table 2), thus lowering the entrainment rate of air from above. Cloud LWP increases in the absence of notable dry-air entrainment, allowing for stronger cloud top LW radiative cooling and subsequent precipitation development within cloud. As a result, BL temperatures are cooler with imposed subsidence than without (Fig. 3**i**), due to the combined effect of reduced entrainment, strong cloud top radiative cooling and enhanced evaporative cooling below cloud. Additionally, the sub-cloud layer becomes more moist and well-mixed with increasing levels of subsidence (Fig. 3**a–c**) as below-cloud rain evaporation generates TKE and promotes convective overturning in the BL. These findings are consistent with observations of precipitating pockets of open cells (POCs), where rain evaporation below cloud was found to cool and moisten the BL (vanZanten and Stevens, 2005).

A moist layer is maintained close to the surface in the CNTRL simulations (Fig. 3**a**), below the sub-cloud mixed layer, whereas this moisture source is eroded in the subsidence cases. Additionally, the CNTRL cases present a minor BL $\Theta_{il}$ inversion, and a stronger $Q_{tot}$ inversion, at approximately 500 m. The combination of these inversions and the moist surface layer suggests that the CNTRL simulations are, in fact, more strongly decoupled from the surface than the subsidence cases at the time step shown (9 h, Fig. 3). However, the subsidence cases display a similar strongly decoupled profile in TKE as the CNTRLs at earlier times (e.g. 5 h, Fig. 3). TKE increases with time in the BL when subsidence is imposed, and appears to promote top-down mixing of TKE through the sub-cloud layer towards the surface by the end of the simulations, tending towards a coupled profile. With more convection caused by strong rain evaporation below cloud, more BL mixing occurs. However, cloud top TKE still dominates the BL profiles in the LO- and HISUB cases, suggesting that mixing throughout the BL is still not homogeneous and the clouds remain approximately decoupled from the surface by the termination time of the simulations.

When consistent surface temperatures and large-scale subsidence are modelled, $W_{sub}$ acts to promote convection through heightened TKE at cloud top and strong evaporation below cloud. This effect appears to be linearly-related to the magnitude of $W_{sub}$. The opposite effect occurs when a combination of subsidence and a warming surface is imposed: higher levels of subsidence act to stabilise the Sc layer and suppress the formation of cumuli from the warming surface (as is seen in the CNTRL_SURFWARM case). TKE production is positively influenced in the CNTRL and LOSUB_SURFWARM cases, with strongly separated cloud and surface sources, and peak values approximately three times greater than modelled in test 1 (Tables 2 and 3). $w'\Theta'$ and $w'Q_{vap}'$ are significantly larger below cloud in CNTRL_SURFWARM at the time step shown in Fig. 9**(h,i)** due to the formation of the below-cloud cumuli; these do not form in HISUB_SURFWARM, and far fewer form





in LOSUB_SURFWARM. Cloud top TKE splits in two in both CNTRL and LOSUB_SURFWARM (Fig. 9**a,b**); however, it is unlikely that this is a resolution artefact as the vertical resolution is consistent through this altitude range. It is possible that the PW advection scheme is introducing spurious oscillations into the advected quantities, caused by the the sharp gradient at the cloud boundary (as discussed by Gray et al., 2001), due to the formation of these dynamic cumuli. Peak TKE is only

marginally stronger in HISUB_SURFWARM than in test 1 (Tables 2 and 3), suggesting that the higher level subsidence offsets efficient in-cloud TKE production when the system is additionally forced by a warming surface.

The gradual coupling of TKE sources seen in HISUB_SURFWARM is likely influenced by the strong evaporative cooling below cloud, which acts to offset the two sources of strong heat and moisture fluxes and make the coupling process more stable. By suppressing the formation of below-cloud cumuli, subsidence acts to produce a stable, yet dynamic, Sc layer, whilst strong

convection and spatial heterogeneity are simulated with low or no subsidence. With more heterogeneity, there is an increased likelihood for instability in the cloud layer, which will likely influence the fate of the cloud downstream.

### 4.4  Role of domain resolution

Whilst CAOs are discussed to motivate our study, we must stress that our chosen domain configuration is not optimal for the explicit study of Sc-to-cumulus transitions downstream in a CAO. Large, high resolution domains are required to accurately

resolve the small-scale microphysical processes within these phenomena (Field et al., 2014); however, our domain size and resolution are restricted by computational expense. Bretherton et al. (1999) demonstrated that our spatial resolution may allow entrainment rates to be overpredicted by approximately 50 % (Connolly et al., 2013). Whilst the authors concluded that the resolutions imposed here can still provide a useful insight into BL evolution, accurately-resolved turbulence requires a higher model resolution. Feingold et al. (2015) found that a higher-resolution setup produces enhanced BL convection and a deeper

BL depth. By increasing spatial resolution, Wang and Feingold (2009a) found that the simulated vertical mixing of vapour and $\Theta$ fields improved, and the modelled LWP increased, in their open cellular convection simulations.

To test the influence of resolution on our findings, we increase the horizontal resolution to 60 m and the vertical resolution to 10 m, whilst maintaining the domain height. This setup therefore decreases the spatial extent of the domain by half. Vertical resolution was 10 m up to 2000 m, decreasing to 20 m above this height. By increasing our model resolution, we aim to provide

a more accurate representation of the modelled entrainment rates. Due to computational expense, only two test cases are considered; the CNTRL and HISUB simulations from test 1 (Sec. 3.1).

The influence of increasing model resolution on the LWP and IWP in the CNTRL setup is shown by the black and the grey traces in Figs. 12**(a,b)**. Little difference between the domain-averaged LWP and IWP can be identified between these CNTRL cases. In the HISUB example – where the higher resolution is shown in pink and the default in red – increasing the model

resolution amplifies the irregularities in both the LWP and IWP traces. In particular, the IWP is significantly more variable with time than the CNTRL setup. Adding a high level of large-scale subsidence and increasing the model resolution allows for more vigorous convective activity to develop with comparison to our CNTRL simulations.

In general, increasing the resolution does not alter the trends identified previously; for example, the positive below-cloud moisture fluxes, higher below-cloud rain evaporation rates, and increased TKE with increasing $W_{sub}$. In fact, it should be noted




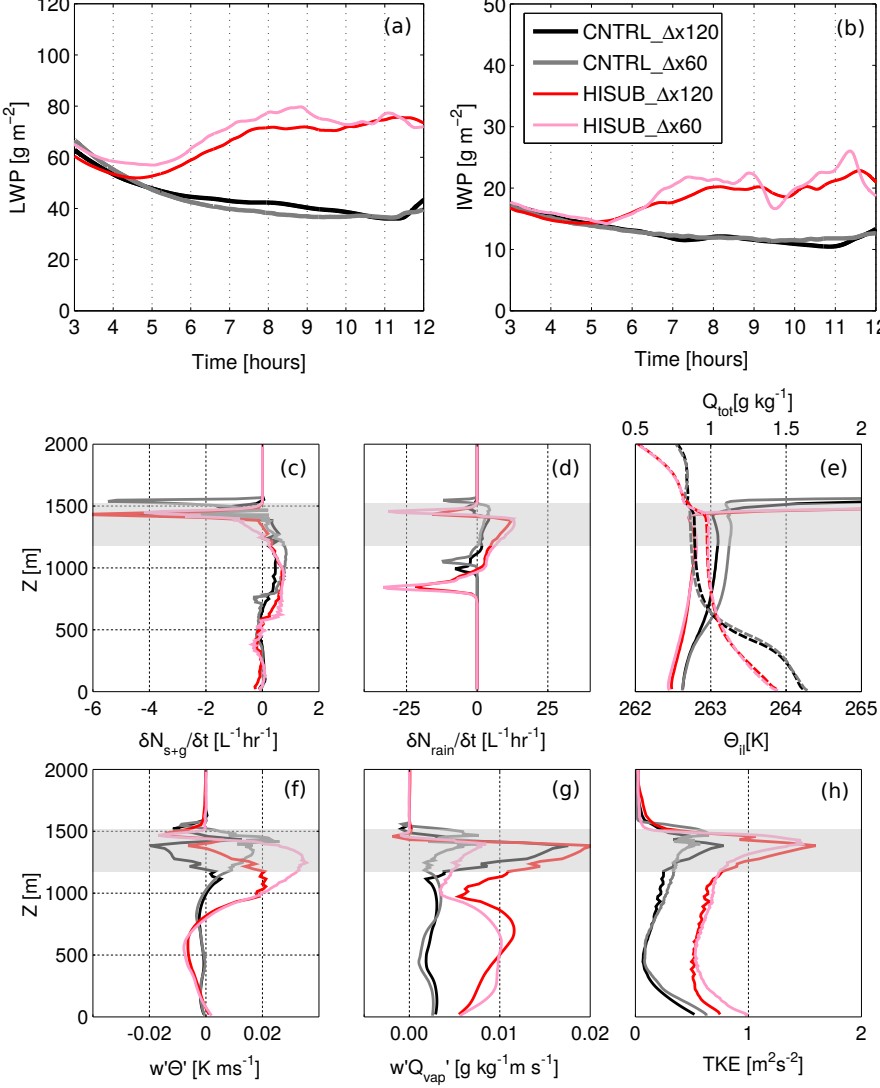

**Figure 12.** Influence of domain resolution on changing imposed large-scale subsidence. Only the CNTRL and HISUB cases are considered. LWP **(a)** and IWP **(b)** time series for simulations with 120 m resolution (default configuration) and 60 m resolution (high resolution configuration). **Black:** CNTRL, default, **grey:** CNTRL, high resolution, **red:** HISUB, default, and **pink:** HISUB, high resolution. **c–h:** Vertical profiles (at 9 h) of **(c):** solid precipitation (snow + graupel) tendency, **(d):** rain tendency ($\delta N_{rain}/\delta t$), **(e):** ice-liquid potential temperature ($\Theta_{il}$, solid) and total water mixing ratio ($Q_{tot}$, dashed), **(f):** buoyancy flux (w'$\Theta$'), **(g):** vertical flux of water vapour (w'$Q_{vap}$') and **(h):** total turbulent kinetic energy (TKE). Fluxes shown are total (sub-grid + advected) time-averaged quantities. Area in grey represents CNTRL_$\Delta$x120m cloudy regions.

that the below cloud rain evaporation rates are enhanced with comparison to the coarse resolution HISUB case, suggesting that





the evaporation rates shown in Sects. 3.2, 3.3, and 3.4 may be underestimated. Increased rain evaporation rates at cloud top may be influencing the snow sublimation rates: with more rain evaporating, the humidity may be maintained above the ice supersaturation threshold, thus suppressing the rate at which snow sublimes. Increasing the vertical resolution allows a greater peak $N_{rain}$ to be modelled, whilst little difference can be identified with $N_{s+g}$ (not shown, Fig. S11); therefore, there is more

rain available to evaporate, sustaining the humidity above ice supersaturation. The $Q_{tot}$ profiles illustrate clear decoupling in the CNTRLs, with a weaker inversion in the HISUB cases. Additionally, both $\Delta x60$ simulations produce a greater TKE peak towards the surface, in addition to the peak simulated at cloud top, due to the dominating influence of the sub-grid contribution to the TKE towards the surface.

Whilst we can test the influence of increased resolution on our findings, increasing our domain size would be too compu-

tationally expensive for our setup. Larger domains are often used to allow mesoscale interactions between developing open convective cells to be resolved. Schröter et al. (2005) suggest that a domain of $100 \times 100$ km, with 50–100 m spatial resolution, is required to truly encapsulate any mesoscale interactions between developing convective cells in CAOs. We cannot speculate what mesoscale interactions may occur between the different scenarios presented here; however, one must note that such interactions have been previously simulated to occur over the transition between closed and open convective cells in CAOs, thus

these effects should be investigated in further work.

## 4.5  Broader implications

Increased convection is modelled within mixed-phase BL Sc with increased subsidence, driven by radiative cooling at cloud top and rain evaporative cooling at cloud base. By enforcing the BL temperature inversion, subsidence reduces entrainment rates from above and thus allows for a greater LWP (and often, IWP) and efficient precipitation development. With more

precipitation evaporating below cloud – coupled with efficient LW radiative cooling at cloud top – the cloud layers become more convective, with increased TKE throughout the BL. These dynamic clouds will be better sustained against the WBF mechanism. This is a crucial result for the understanding of mixed-phase Sc in the Arctic – particularly in the Arctic spring – where high pressure, stable conditions dominate across the region. These clouds have been observed to persist for long periods of time, and subsidence caused by large-scale meteorology could be acting to sustain these clouds microphysically

against dissipation or glaciation. Kay and Gettelman (2009) found lower cloud fractions in high pressure regions; however, it is important to note that this study considered the high Arctic, where the surface was ice-covered. Our results indicate a microphysical sensitivity to subsiding air associated with high pressure systems in the ocean-exposed low-, or sub-, Arctic regions; regions which commonly experience CAOs.

## 5  Conclusions

Large-scale subsidence is often imposed in LES models as a tuning factor to maintain cloud top height; however, the influence of this parameter on mixed-phase cloud microphysics has not been previously investigated. Here, we have shown how large-scale subsidence affects the microphysical structure of BL mixed-phase Sc using the UK Met Office Large Eddy Model





(LEM, UK Met Office, Gray et al., 2001). By subjecting four idealised scenarios – a stable Sc, varied droplet ($N_{drop}$) or ice ($N_{ice}$) number concentrations, and a warming surface – to different levels of subsidence, we have identified a clear relationship between subsidence and convection development, with potential implications for mixed-phase BL clouds forming in the ocean-exposed low-, or sub-, Arctic regions.

Key features identified in this study are as follows:

– With no surface forcing (tests 1–3), increasing the imposed large-scale subsidence ($W_{sub}$) reinforces the BL temperature inversion and thus reduces entrainment from the free troposphere. With less dry air from aloft mixing into the clouds, a greater LWP (and often, IWP) develops, allowing for efficient cloud top radiative cooling and downdraught production. Precipitation formation is enhanced in these downdraught regions, and all of the rain produced evaporates below cloud.

The combination of strong cloud top radiative and below-cloud evaporative cooling generates more TKE within the BL, leading to enhanced turbulent overturning throughout the cloud layer, positively-forcing the LWP. These linked processes combine to form a feedback loop consisting of $W_{sub}$, LWP, rain evaporation, and TKE development.

– Imposed large-scale subsidence has a greater impact on the LWP and IWP than the chosen microphysical changes; varying $N_{drop}$ (Fig. 4) or $N_{ice}$ (Fig. 6). BL TKE, $w'^2$, and cloud LWP increase with increasing $W_{sub}$, suggesting that the

clouds may be more robust against dissipation or glaciation via the WBF mechanism. Quiescent, less dynamic clouds are modelled under no subsidence; clouds which may be more vulnerable to the WBF mechanism.

– In microphysical scenarios which promote efficient rain production (low $N_{drop}$ or low $N_{ice}$), $W_{sub}$ enhances rain production and evaporation rates, TKE at cloud top and at the surface, and turbulent activity throughout the BL. Modelled $N_{rain}$ increases with $W_{sub}$, whilst snow number concentrations marginally decrease. Modelled rain evaporates more efficiently

than snow, and stimulates the cloud dynamically by introducing perturbations in moisture and temperature below cloud. Only precipitation as snow reaches the surface, mirroring observations of marine mixed-phase Sc in the Arctic (Young et al., 2016) and in CAOs.

– Subsidence affects both the liquid and ice phases when properties related to the liquid phase are altered (test 2, Fig. 4). However, altering the ice phase feeds back onto the liquid phase through the influence of the WBF mechanism (test

3, Fig. 6). Clouds with greater ice number concentrations suppress the liquid phase; therefore, $N_{ice}$ has a key role in mediating the strength of turbulent overturning induced in these mixed-phase clouds. Subsidence readily affects the concentration of rain produced through convective overturning, whilst the ice phase is relatively insensitive to these changes. With more dynamical motion in the modelled cloud, the liquid phase may be sustained more effectively against the WBF mechanism.

– In the absence of surface warming, all modelled BLs display a stable $\Theta_{il}$ profile; however, cloud sources of TKE, moisture, and heat are decoupled from the surface due to strong below-cloud rain evaporation. This decoupling allows radiative cooling at cloud top and evaporative cooling below cloud to drive convective activity in the cloud layers,





irrespective of surface sources. The HISUB simulation tends towards a coupled, well-mixed BL through the top-down and bottom-up propagation of TKE (test 1, Fig. 3**c**).

– The feedbacks identified from test 1–3 are not so clearly related when a warming surface is additionally imposed: significantly larger values of w'$Q_{vap}$' and w'$\Theta$' are modelled with no $W_{sub}$, coinciding with the rapid BL coupling shown in Fig. 9**(a)**. In-cloud rain production rates produced in CNTRL_SURFWARM are also much greater than modelled without surface forcing in test 1. A warming surface, and a lack of subsidence, acts to dynamically stimulate the modelled cloud from below, similar to how subsidence stimulates it from above.

– Below-cloud cumuli form in CNTRL_SURFWARM, and to a lesser extent in LOSUB_SURFWARM, which act to push cloud top higher, generate high LWPs, and cause significant spatial heterogeneity in the cloud layer. This cumuli formation is suppressed when under high levels of subsidence (HISUB_SURFWARM); the combination of these two forcings counteract one another to produce a stable, yet dynamic, Sc layer.

– In all subsidence cases, the $\Theta_{il}$ profiles become unstable towards the warm surface. The CNTRL_SURFWARM case couples strongly at 10 h (Fig. 9**a,b**), whilst the dominating cloud sources of TKE in HISUB_SURFWARM allows the cloud to couple more gradually to surface TKE, moisture, and heat sources. The gradual coupling of HISUB_SURFWARM is likely influenced by the strong evaporative cooling below cloud (test 6, Fig. 9**c**).

– Similar to our coarse resolution simulations, more in-cloud and surface TKE is modelled in HISUB_$\Delta$x60 than in CNTRL_$\Delta$x60. Increasing model resolution exaggerates the effect of imposing large-scale subsidence. Below-cloud rain evaporation rates and in-cloud w'$\Theta$' increase with increasing resolution, whilst BL $\Theta_{il}$ and TKE are largely unaffected.

This study presents a clear relationship between large-scale subsidence and the development of convection in liquid-dominated mixed-phase clouds common to the sub-Arctic. We propose that the influence of large-scale subsidence in both sub-Arctic CAOs and Arctic mixed-phase Sc should be considered in further work, with models of different scales. In particular, it would be beneficial to study the development of CAO flows – with a high-resolution, large domain – under a transitional profile of subsidence; i.e. flowing from a high pressure region. Our results suggest that a high $W_{sub}$ will amplify turbulent activity and rain production/evaporation in any stable mixed-phase Sc modelled, and a weakening of subsidence alongside a warming surface will likely promote cloud top ascent, below-cloud cumuli formation, and strong spatial heterogeneities throughout the cloud layer. Therefore, further investigating the role of subsidence in CAO flows will be beneficial to our ability to accurately model and understand the break up of these cloud decks. More generally, comprehending the physical impact of subsidence on marine mixed-phase cloud microphysics at higher latitudes will allow us to better predict how clouds in the Arctic region may change in the depleted sea ice future.





# 6 Code availability

Please contact the UK Met Office for LEM code requests.

# 7 Data availability

LEM model runs are archived at the University of Manchester and are available on request.

5 *Competing interests.* The authors declare that they have no conflict of interest.

*Acknowledgements.* GY was funded by the National Environment Research Council (NERC) as part of the ACCACIA campaign (grant NE/I028696/1). PJC acknowledges funding from EU FP7 project BACCHUS (grant agreement no. 603445).





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
