# Peer review of "Large-scale subsidence promotes convection in sub-Arctic mixed-phase stratocumulus via enhanced below-cloud rain evaporation"

_Atmospheric Chemistry and Physics, 2017_

## Referee Comment (RC1) · Anonymous Referee #1 · 8 Sep 2017

Review of "Large-scale subsidence promotes convection in sub-Arctic mixed-phase stratocumulus via enhanced below-cloud rain evaporation" by Gillian Young et al.

This study presents a very nice series of simulations to test the response of Arctic mixed-phase clouds to subsidence under several different scenarios. This is a very little studied topic for these clouds, and the topic is appropriate for ACP. The authors do a good job of presenting not just the results, but in providing in depth discussion for why the changes occur. However, I have questions about some of their process arguments, and the paper overall needs to be edited substantially for clarity and be made more concise. I recommend major revisions.

[Figure]

Major Comments:

1. This is an extremely long paper, by my estimate 10-11 thousand words. I appreciate that there are several sets of simulations to discuss, but I still found that the paper was very repetitive at times and the writing was not always clear or well organized. I think that it could be substantially shortened without removing any of the main points. I've pointed out several specific instances where improvements could be made below.

2. Page 13, Line 8. It seems that the authors have misread the plot. Altering Nice has a much larger impact than changing Wsub, not the other way around. This false interpretation is repeated in the conclusions on Page 27, Line 14. This is also an important point for understanding my next comment.

3. The primary hypothesis is that increased subsidence retards dry air entrainment, leading to higher LWP and increased rain formation. The former allows for greater cloud top radiative cooling while the latter allows for greater sub-cloud evaporation and turbulence production. My question though is why do you not see a similar response when decreasing Nice? When decreasing Nice, you have much higher LWP, more rain production and sub-cloud evaporation, but you do not seem to get much change to TKE. Some differences exist, but they are not nearly as large as the differences due to varying Wsub, even though the change in LWP is larger when varying Nice. Why do we not see a similar response?

4. It is odd to me that the authors consistently show dNrain/dt to talk about increased/decreased evaporation and not dqrain/dt (rate of change of rain mass). Just because there are more/fewer drops being evaporated doesn't necessarily mean that more/less rain mass is being evaporated. And it is the amount of mass that controls the latent cooling magnitude and feeds into turbulence. Showing rain mass and rain mass rates of change instead would help to strengthen their arguments. The same comment applies to snow sublimation.

Minor Comments:

5. The title doesn't seem to reflect the content of the paper well. The below-cloud evaporation is only given as one contributing factor to the promotion of convection in these clouds. Also, it is only one aspect of the subsidence issue among many that are discussed in the text.

6. The introduction has lots of good information, but I think that it is confusing sometimes about whether the results pertain to the subtropics, Arctic, or both. Also, I find the motivation for the study a little confusing in the last paragraph of the introduction. The focus is on CAO transitions, but most of the study is not focused on CAOs. Is decreasing subsidence associated with CAO transitions? If so, this has not been clearly stated, and the link to tests 1-3 is not clearly made later.

7. Page 3, Lines 13-16. So cloudiness and high pressure are correlated in subtropical marine environments, and anti-correlated in the Arctic? Why?

8. Page 5. The text describes tests 1, 2, and 4, but not test 3. The description of the control simulation should probably be given before describing the tests.

9. Page 7, Lines 27-28. Why do non-zero snow rates implicitly suggest heterogeneity in the snow field?

10. Figure 3. I can't tell which lines are dashed in Fig. 3f (although it's easy enough to figure out).

11. Page 9, 1st paragraph. Why higher LWP? The authors mention later that it is reduced entrainment of dry air, but that could be explicitly mentioned here.

12. Page 9, Lines 21-23. While I certainly agree that each individual droplet will be larger, I don't see why that necessarily means that the LWP must increase. In fact, I would probably expect the opposite response. For lower Ndrop, that you would get more rain production, fallout and evaporation leading to overall reduced LWP.

13. Page 10, 1st paragraph. The profiles of turbulent quantities seem almost unchanged with changing Ndrop, and the differences described are hard to see.

14. Page 11, Line 3. Why would the downdrafts facilitate precipitation production? I primarily associate downdrafts with liquid evaporation and reduction of precipitation.

15. Page 11, Line 14. How is Ndrop decreased? Ndrop is held constant in the simulations.

16. Page 11, Line 17. Smaller effect on Nrain than what?

17. Page 13, Lines 9-10. This sentence is confusing. Please rephrase.

18. Page 13, Line 12. More exaggerated than what? The CNTRL case?

19. Page 15, Lines 5-9. This seems like a minor detail that doesn't need to be discussed. Plus, the trends at 9hrs can't be used to understand how you arrived at the current state at 9hrs.

20. Page 18, Line 1. Increased snow sublimation compared to what?

21. Page 18, Line 13. Incorrect units on TKE.

22. Page 18, Lines 19-20. The discussion is repeating itself.

23. Page 19, Lines 3-5. This sentence is confusing. Please rephrase.

24. Page 21, Line 4. Cloud extent has never been shown. Or do you mean vertical extent? I had interpreted it as cloud fraction. I don't understand how the next sentence is a logical conclusion from this sentence.

25. Page 22, Lines 10-25. If the focus on this section is subsidence and microphysics, then these lines are not necessary.

26. Page 26, Section 4.5. I'm not sure what this section adds to the manuscript. All of the points seem to have been made already.

27. Page 27, Line 9. The authors have not shown that precipitation formation is enhanced in downdrafts.

28. Page 27, Line 12. Wsub cannot possibly be in a feedback loop since it is held constant in the simulations.

29. Page 27, 3rd and 4th points. These points seem to mostly restate the first two conclusion points. In general, I think that the paper could be strengthened by highlighting just three or four main take-home points rather than nine.

---

## Referee Comment (RC2) · Anonymous Referee #2 · 15 Sep 2017

This manuscript describes a series of simulations of mixed-phase stratocumulus clouds designed to elucidate the role of large-scale subsidence in maintaining such clouds. The main conclusion is that subsidence enhances droplet evaporation at cloud top and below the cloud base, as well as supporting the cloud top inversion. Collectively, this isolates the cloud from entrainment of dry subsiding air from above, thereby enhancing in-cloud turbulence and promoting longevity. For southward moving mixed-phase Sc, such as during cold air outbreaks, simulations suggest advection over a relatively warmer surface promotes dynamic coupling and evolution of the cloud, but stabilization under high subsidence. The manuscript is well-researched and thorough, and is well-suited for publication in Atmospheric Chemistry and Physics. I recommend that it

be published after a minor revision addressing several comments below.

General Comments:

1) The manuscript is generally well-written, but it is quite long and the preponderance of details distracts from the take-home messages of each section. It is therefore difficult at times to follow. I am hoping that it can be tightened up throughout with the goal of drawing out the key points.

2) The simulations presented appear to be based on a case study from the eastern Arctic outlined by Young et al. (2017). Terms "Arctic", "low Arctic" and "sub Arctic" are variously used and I find myself somewhat lost geographically. I feel the necessary context may lie in Fig. 2 from Young et al (2017), but it is also not clear how much of the present study is hypothetical or how closely it relates to the previous work.

3) Following on from (2), the conclusions of the study are highly generalized, which is consistent with the experimental design of the simulations, except for the fact that it is ultimately based on a single atmospheric state case at initialization, which the reader learns little about. The importance of this limitation is not clear.

4) I don't understand how the model treats the surface properties and coupling, and thus to what degree dynamic coupling with the surface can feed back to the cloud, or if this can be evaluated at all (e.g., test 1 and test 4).

Specific Comments:

Title: "Large-scale... via ...evaporation" and also enhanced cloud-top radiative cooling, right?

Abstract Line 20: Clarify "warming surface", which you do not mean to be climatological, but rather southward advection.

Page7 Line5: "an" should be "a"

Page9 Line9: For the cases that become dynamically coupled, is the surface becoming

a moisture source?

Page13 Lines1-6: Why the later, more rapid increase in CNTRLD10x2? Is this important somehow to understand the main thesis of this simulation?

Page13 Line5: "earlier" not "more quickly"?

Page13 Line7/Page27 Line13-14: This doesn't seem right. Looks like the LWP response to Nice is much larger than the response to subsidence.

Page15 Line23: Is the ascent of the cloud exacerbating the difference relative to CNRTL in 7B(a,d) since its spatial position is changing relative to CNTRL?

Page21 Line4: Replace "extent" with "depth" or "physical thickness" so as not to be confused with horizontal extent.

---

## Author Comment (AC1) · 22 Nov 2017

**We would like to thank the reviewers for their useful comments and suggestions which have helped us to improve the manuscript.**

**Reviewers' comments**

Reviewer 1, Reviewer 2

**Authors' response is shown in black and bulleted.**
**Quotes from the manuscript are in italics.**

**Please note:** Some figure numbers have been changed in the updated version of the manuscript. New figure numbers are referenced in any related comments. Quoted line numbers are from the revised manuscript.
* * *
This study presents a very nice series of simulations to test the response of Arctic mixed-phase clouds to subsidence under several different scenarios. This is a very little studied topic for these clouds, and the topic is appropriate for ACP. The authors do a good job of presenting not just the results, but in providing in depth discussion for why the changes occur. However, I have questions about some of their process arguments, and the paper overall needs to be edited substantially for clarity and be made more concise. I recommend major revisions.

Major Comments:

1. This is an extremely long paper, by my estimate 10-11 thousand words. I appreciate that there are several sets of simulations to discuss, but I still found that the paper was very repetitive at times and the writing was not always clear or well organized. I think that it could be substantially shortened without removing any of the main points. I've pointed out several specific instances where improvements could be made below.

- We have scaled back the manuscript by about 3 pages (approximately 2200 words) following Reviewer 1's point. Additionally, we have endeavoured to remove as much repetition as we can to clarify the main points being made. Instead of re-iterating the findings of each test in the subsequent one, we have focused more on what is new in that chosen scenario. We have also ensured that no comparisons between test cases are made in the Results section; these have now been moved to the Discussion.

2. Page 13, Line 8. It seems that the authors have misread the plot. Altering Nice has a much larger impact than changing Wsub, not the other way around. This false interpretation is repeated in the conclusions on Page 27, Line 14. This is also an important point for understanding my next comment.

- This is correct, and was also highlighted by Reviewer 2. We did inaccurately describe the plot and have rectified this in the manuscript (now page 12, line

1). This paragraph now begins with: "*Trios can be easily identified in Fig. 6(a)...*". We apologise for any confusion caused.

3. The primary hypothesis is that increased subsidence retards dry air entrainment, leading to higher LWP and increased rain formation. The former allows for greater cloud top radiative cooling while the latter allows for greater sub-cloud evaporation and turbulence production. My question though is why do you not see a similar response when decreasing Nice? When decreasing Nice, you have much higher LWP, more rain production and sub-cloud evaporation, but you do not seem to get much change to TKE. Some differences exist, but they are not nearly as large as the differences due to varying Wsub, even though the change in LWP is larger when varying Nice. Why do we not see a similar response?

- This effect can be explained by considering the $\delta Q_{sg}/\delta t$ values in each of these cases. As such, a figure showing $\delta Q_{sg}/\delta t$ and $\delta Q_{rain}/\delta t$ has been added as Fig. 8 to show the differences in tendencies between CNTRL_D10, CNTRL_D10x0.5, and CNTRL_D10x2. The following discussion has been added to the manuscript at page 14, lines 5-15:

  "*LWP and below-cloud rain evaporation are enhanced in CNTRL_D10x0.5 with comparison to CNTRL_D10 and CNTRL_D10x2; however, $w'^2$ is not strongly affected (Fig. 6). Figure 8 shows $\delta Q_{sg}/\delta t$ and $\delta Q_{rain}/\delta t$ at 9 h to illustrate differences between the D10×0.5, D10, and D10×2 CNTRL cases. $\delta Q_{sg}/\delta t$ is similar in the D10 and D10x0.5 simulations, whilst the LWP and rain evaporation/production processes are positively-forced by decreasing $N_{ice}$. In the turbulent subsidence cases, $\delta Q_{sg}/\delta t$ does increase below cloud with increasing $W_{sub}$ (Fig. 7A). This is the only key difference between decreasing $N_{ice}$ and increasing $W_{sub}$; therefore, increased latent heating through snow growth at cloud base – alongside heightened below-cloud rain evaporation and efficient cloud-top radiative cooling via a high LWP – is required to generate the heightened TKE (as illustrated here by $w'^2$) in these scenarios. Convection is suitably induced in LO- and HISUB_D10x0.5 as the modelled snow growth rates are greater (Fig. S8). Whilst the same $N_{ice}$ is modelled in each of these scenarios, the subsidence cases produce a much colder BL than CNTRL_D10x0.5; therefore, the environmental conditions in LO- and HISUB_D10x0.5 facilitate snow growth below cloud, whilst the control produces comparatively inefficient growth conditions.*"

4. It is odd to me that the authors consistently show dNrain/dt to talk about increased/decreased evaporation and not dqrain/dt (rate of change of rain mass). Just because there are more/fewer drops being evaporated doesn't necessarily mean that more/less rain mass is being evaporated. And it is the amount of mass that controls the latent cooling magnitude and feeds into turbulence. Showing rain mass and rain mass rates of change instead would help to strengthen their arguments. The same comment applies to snow sublimation.

- We have updated Figures 3, 5, 7, 10 and 13 to show rates of change of mass instead of number. In Figs. 5 and 7, we have kept the number concentrations of snow+graupel and rain as overlaid contours, as we feel this provides a holistic representation of how the precipitable Q-fields are changing under large-scale subsidence. This change made us realise the importance of latent heating due to snow growth at cloud base; therefore, we thank Reviewer 1 for the suggestion.

5. The title doesn't seem to reflect the content of the paper well. The below-cloud evaporation is only given as one contributing factor to the promotion of convection in these clouds. Also, it is only one aspect of the subsidence issue among many that are discussed in the text.

- The title has been changed to make a more general statement about the contents of the study:

  "*Relating large-scale subsidence to convection development in Arctic mixed-phase marine stratocumulus*"

6. The introduction has lots of good information, but I think that it is confusing sometimes about whether the results pertain to the subtropics, Arctic, or both. Also, I find the motivation for the study a little confusing in the last paragraph of the introduction. The focus is on CAO transitions, but most of the study is not focused on CAOs. Is decreasing subsidence associated with CAO transitions? If so, this has not been clearly stated, and the link to tests 1-3 is not clearly made later.

- We had originally included a detailed overview of findings from previous studies which investigated Sc-to-cumulus transitions on a microphysical level to show the current state of knowledge. Given that little work has been done on mixed-phase clouds in a CAO, we showed results from studies of subtropical clouds. Some of these studies have suggested that subsidence may influence stratocumulus to cumulus transitions, and this finding formed part of the motivation for this work. To this end, we felt that details of these studies should be included.

  We do, however, see Reviewer 1's point that this has made the Introduction misleading as we do not simulate a CAO. We cannot use our model to do this (as discussed in Sect. 4.4); we have instead used our model to identify what impact subsidence has on mixed-phase cloud microphysics on a more fundamental level. This investigation may therefore allow some inferences to be made about the role of subsidence in a CAO.

  We have modified the Introduction to reflect Reviewer 1's comments; specifically, we have been clearer on whether we are discussing polar or tropical studies, and the roles of both precipitation and subsidence. We have removed some of the discussion of CAOs and streamlined what remains to make the relevance to our study much clearer. Additionally, we have added a clearer link at the end of the Introduction section (page 3, lines 3-9) as to why we have designed the experiments in this way: to demonstrate the microphysical feedbacks which are affected by subsidence and test how the combination of subsidence and a warming surface can affect BL development.

7. Page 3, Lines 13-16. So cloudiness and high pressure are correlated in subtropical marine environments, and anti-correlated in the Arctic? Why?

- This is correct, and contributed towards the motivation of this study. In subtropical marine environments, high pressure systems have been found to correlate with cloudiness due in part to the presence of a surface heat source; the ocean. Most Arctic studies consider sea ice-covered surfaces, which are devoid of this source. Arctic clouds in high pressure systems are therefore cut off from moisture sources from below (by the sea ice barrier) and above (by the subsidence and strong BL temperature inversion attributed to the high

pressure system), often leading to cloud dissipation and reduced cloud fractions. We use our experiments to show how mixed-phase clouds in Arctic marine environments may be influenced by large-scale subsidence, as this has not previously been considered in such detail. Instead of cloud dissipation, we find that the ocean surface heat source allows the clouds to behave similarly to at lower latitudes. We suggest that the reason for this is the increased inversion strength from the high pressure system (large-scale subsidence), which promotes cloudiness in these scenarios through efficient cloud top radiative cooling, below-cloud rain evaporation/snow growth, and convection development.

8. Page 5. The text describes tests 1, 2, and 4, but not test 3. The description of the control simulation should probably be given before describing the tests.

- We have re-arranged the text following Reviewer 1's comments. The control experiment is now described before the test cases (page 4, line 5-9), and text from Sect. 3.3 has been moved to the following paragraph (page 4, lines 10 - 15) to provide context for test 3.

9. Page 7, Lines 27-28. Why do non-zero snow rates implicitly suggest heterogeneity in the snow field?

- This field is averaged across the domain; therefore, if no snow reached the surface, these rates would be zero at low altitude, or largely negative to indicate significant sublimation across the domain. We realise that this is not particularly clear with the current wording; therefore, we have referenced the Z-X slices shown in the Supplement as these show the heterogeneous distribution of snow much more clearly (page 7, lines 3-5):

  "*Precipitation as snow does reach the surface; however, the spatial distribution becomes more heterogeneous with increased $W_{sub}$ (not shown, Fig. S3 – S5)*"

10. Figure 3. I can't tell which lines are dashed in Fig. 3f (although it's easy enough to figure out).

- We feel that both of these traces are important to show, so instead of removing the dashed $Q_{tot}$ lines we have made them thicker and more distinct for the reader.

11. Page 9, 1st paragraph. Why higher LWP? The authors mention later that it is reduced entrainment of dry air, but that could be explicitly mentioned here.

- We were unclear on which paragraph was being referred to as the first paragraph on Page 9 does refer to the higher LWP. Upon re-reading this section, we thought it may have been the first paragraph of Sect. 3.1 that was being referenced; therefore, we have noted the stronger inversion in this paragraph alongside the initial comment regarding the increased LWP (page 6, lines 3 – 5):

  "*A stable Sc is modelled in the absence of $W_{sub}$ (CNTRL, Fig. 2a). Increasing $W_{sub}$ (LO- and HISUB) makes the temperature inversion stronger, as shown in Table 2, thus reducing entrainment into the cloud from above the BL.*"

12. Page 9, Lines 21-23. While I certainly agree that each individual droplet will be larger, I don't see why that necessarily means that the LWP must increase. In fact, I

would probably expect the opposite response. For lower Ndrop, that you would get more rain production, fallout and evaporation leading to overall reduced LWP.

- Yes, for a lower $N_{drop}$ one would expect more rain production and fallout; however, as all of our modelled rain evaporates below cloud, this moisture is retained by the system and recycled into the cloud. $N_{drop}$ is prescribed and is therefore fixed throughout the simulation, and only the liquid mass is prognostic. Any condensed mass is automatically distributed amongst this fixed number population, and any increase in mass corresponds to an increase in size. Therefore, a lower $N_{drop}$ corresponds to larger droplets. However, rain number and mass concentration are both represented in the model so, whilst the droplet category is limited by number, the rain category is not. This is indeed what we see, as shown by the black contours in Fig. 5B: a greater $N_{rain}$ is modelled with Ndrop50. This increased $N_{rain}$ therefore contributes towards a greater LWP.

  We realise that the inclusion of this comment is misleading without this explanation; therefore, in the interest of clarity, it has been removed.

13. Page 10, 1st paragraph. The profiles of turbulent quantities seem almost unchanged with changing Ndrop, and the differences described are hard to see.

- We agree that these differences are small and difficult to see. We have re-ordered most of this subsection (pages 9-10) to focus less on the small changes previously discussed and more on the key messages we wish to convey. Additionally, we have re-arranged the subsequent sections in a similar manner to emphasise the key differences and not focus so much on the smaller ones.

14. Page 11, Line 3. Why would the downdrafts facilitate precipitation production? I primarily associate downdrafts with liquid evaporation and reduction of precipitation.

- Cloud top longwave cooling would produce the downdraughts consisting of colder air. These colder temperatures promote vapour growth of ice/snow crystals and condensational growth of cloud droplets in the downdraught column throughout the cloud. Additionally, the downwards motion would provide good opportunity for collision-coalescence between droplets, such that they reach precipitable sizes. Liquid evaporation and precipitation reduction would certainly occur (and does occur) toward cloud base if the particles fall below the lifting condensation level; however, we are referring to downdraughts within the cloud layer. Nonetheless, we have removed this comment in the process of tightening up the manuscript.

15. Page 11, Line 14. How is Ndrop decreased? Ndrop is held constant in the simulations.

- $N_{drop}$ is prescribed and is chosen prior to running each simulation. It is this prescribed number concentration that is altered. Once the simulations are initiated, $N_{drop}$ remains constant.

16. Page 11, Line 17. Smaller effect on Nrain than what?

- A smaller effect than decreasing $N_{drop}$. We have clarified this point in the manuscript (page 11, line 11):

  "*Increasing $N_{drop}$ has a smaller effect on $N_{rain}$ than decreasing it, as expected by the...*"

17. Page 13, Lines 9-10. This sentence is confusing. Please rephrase.

- We agree that this sentence was confusing; however, we decided to remove it entirely rather than rephrasing as the point being made was extremely minor.

18. Page 13, Line 12. More exaggerated than what? The CNTRL case?

- Correct, we have clarified this in the manuscript (page 12, lines 5-6).

  *"… for example, the extremes in the w'Θ' profiles are more exaggerated in the LO- and HISUB cases than the CNTRL when a lower $N_{ice}$ is modelled… "*

19. Page 15, Lines 5-9. This seems like a minor detail that doesn't need to be discussed. Plus, the trends at 9hrs can't be used to understand how you arrived at the current state at 9hrs.

- We agree that this is a minor detail which has been given too much attention. In the interest of tightening up the manuscript, we have removed this segment of text, and Fig. S9.

20. Page 18, Line 1. Increased snow sublimation compared to what?

- We were referring to the fact that the $\delta N_{sg}/\delta t$ rates shown were negative <750 m in the CNTRL_SURFWARM case and <500m in the subsidence cases, suggesting that the snow was subliming more so as it reaches the surface in this case than without surface warming. This segment of text has now been removed due to changing $\delta N_{sg}/\delta t$ to $\delta Q_{sg}/\delta t$ throughout the manuscript.

21. Page 18, Line 13. Incorrect units on TKE.

- This has been corrected to $m^2 s^{-2}$ (page 17, line 16)

22. Page 18, Lines 19-20. The discussion is repeating itself.

- This discussion has been re-located to earlier in the section; however, the ordering of this section has changed substantially on revision and therefore some re-wording has also been carried out.

23. Page 19, Lines 3-5. This sentence is confusing. Please rephrase.

- We have rephrased this sentence as requested (now page 20, lines 32-33), and added a reference to a similar figure of the CNTRL simulation included in the Supplement:

  *"Similarly, total ice number concentrations (ice+snow+graupel, $N_{isg}$, Fig. 11b) are largely unaffected by a warming surface (with comparison to Fig. S3b); however, both $Q_{liq}$ and $N_{rain}$ increase. "*

24. Page 21, Line 4. Cloud extent has never been shown. Or do you mean vertical extent? I had interpreted it as cloud fraction. I don't understand how the next sentence is a logical conclusion from this sentence.

- Reviewer 2 also raised this issue; we apologise for any confusion. We were referring to the cloud depth and indeed made a poor choice of wording. We have changed this to "cloud depth" in the manuscript (now page 19, line 6).

25. Page 22, Lines 10-25. If the focus on this section is subsidence and microphysics, then these lines are not necessary.

- We agree with Reviewer 1's comment: lines 10-17 have been moved to Sect. 4.3, whilst lines 18-25 have been reworded into Sect. 3.1. Some of the discussion has been reworded and reworked into existing paragraphs on revision.

26. Page 26, Section 4.5. I'm not sure what this section adds to the manuscript. All of the points seem to have been made already.

- We had included this section to summarise the main points as the paper is quite detailed. We understand the reviewers concern that it is just repetition; therefore, we have removed this and ensured that the points discussed are included in the Conclusions section.

27. Page 27, Line 9. The authors have not shown that precipitation formation is enhanced in downdrafts.

- The Z-X cross sections (Figs. 11, 12) show $N_{rain}$ as white contours on the second panel and W as shading on the bottom panel. Higher $N_{rain}$ within the cloud layer does appear to be co-located with strong downdraughts (this is clear in Fig. 12). This was also inferred by theory, as downdraughts would create conditions for efficient collision-coalescence, allowing the droplets to grow to precipitable sizes. We have removed this statement from the Conclusions and moved it to the Discussion section (Sect. 4.2, page 20, lines 33-34), and added a reference to these figures in the manuscript as explanation of this fact.

28. Page 27, Line 12. Wsub cannot possibly be in a feedback loop since it is held constant in the simulations.

- We agree that this point was incorrectly made: it was our intention to state that a feedback loop between the LWP, rain evaporation, snow growth, and TKE development was positively forced by imposing greater levels of $W_{sub}$. Yes, $W_{sub}$ is not part of this loop, but it makes this loop stronger. This was our intention, and we have re-worded the manuscript to reflect this (page 24, lines 27 - 30):

  *"The combination of strong cloud top radiative cooling, below-cloud evaporative cooling, and latent heating from snow growth at cloud base generates more TKE within the BL. These three requirements combine to form a feedback loop consisting of LWP, below-cloud rain evaporation/snow growth, and TKE development, positively forced by the magnitude of $W_{sub}$."*

29. Page 27, 3rd and 4th points. These points seem to mostly restate the first two conclusion points. In general, I think that the paper could be strengthened by highlightingjust three or four main take-home points rather than nine.

- We agree in hindsight it was not helpful to include so many points in the Conclusion. As suggested by Reviewer 1, we have scaled this back to 4 key "take home" messages that we wish to emphasise to the reader.

This manuscript describes a series of simulations of mixed-phase stratocumulus clouds designed to elucidate the role of large-scale subsidence in maintaining such clouds. The main conclusion is that subsidence enhances droplet evaporation at cloud top and below the cloud base, as well as supporting the cloud top inversion. Collectively, this isolates the cloud from entrainment of dry subsiding air from above, thereby enhancing in-cloud turbulence and promoting longevity. For southward moving mixed-phase Sc, such as during cold air outbreaks, simulations suggest advection over a relatively warmer surface promotes dynamic coupling and evolution of the cloud, but stabilization under high subsidence. The manuscript is well-researched and thorough, and is well-suited for publication in Atmospheric Chemistry and Physics. I recommend that it be published after a minor revision addressing several comments below.

General Comments:

1) The manuscript is generally well-written, but it is quite long and the preponderance of details distracts from the take-home messages of each section. It is therefore difficult at times to follow. I am hoping that it can be tightened up throughout with the goal of drawing out the key points.

- Reviewer 1 also made this comment: we have tightened up the manuscript significantly following both Reviewers' comments.

2) The simulations presented appear to be based on a case study from the eastern Arctic outlined by Young et al. (2017). Terms "Arctic", "low Arctic" and "sub Arctic" are variously used and I find myself somewhat lost geographically. I feel the necessary context may lie in Fig. 2 from Young et al (2017), but it is also not clear how much of the present study is hypothetical or how closely it relates to the previous work.

- We understand that context is difficult without consulting Young et al., 2017. We use the term "Arctic" for Arctic-wide discussions and "high Arctic" for high (>80°N) latitudes typically covered in sea-ice. We have removed "sub-Arctic" and "low-Arctic" from the discussion to avoid confusion. Most of these terms are used in the literature; however, we have added these latitude ranges to the manuscript to make these locations more obvious to the reader (page 2, line 29). Additionally, we have included more in the Methods section (page 5, lines 6 - 7) relating to the setup of the model:

  "..., centred on 71° in the European Arctic to allow appropriate SW radiation estimates to be calculated by the model."

3) Following on from (2), the conclusions of the study are highly generalized, which is consistent with the experimental design of the simulations, except for the fact that it is ultimately based on a single atmospheric state case at initialization, which the reader learns little about. The importance of this limitation is not clear.

- Given computational expense, we could only conduct our chosen experiments for one set of initial conditions. We understand that variation from these conditions would have been beneficial to appreciating how general our findings are. We are planning on conducting more experiments in the future; specifically, we would like to know how a neutral BL, instead of a stable one, would affect our results. Until we do more work on this, we cannot answer how

important this limitation is. Therefore, we have added more into the Discussion section to make this limitation clearer to the reader (page 21, lines 20-22).

*"... This stability is likely influenced by the stable conditions used to initialise the model, and one must note that only a single set of initial conditions were used in this study."*

4) I don't understand how the model treats the surface properties and coupling, and thus to what degree dynamic coupling with the surface can feed back to the cloud, or if this can be evaluated at all (e.g., test 1 and test 4).

- The LEM computes surface fluxes from the lowest levels of the input fields (e.g. $\Theta$, $Q_{vap}$). Changes in these fields through, for example, evaporative cooling from precipitation, can then be fed back into the surface levels; however, the changes are very small and the surface temperature can be assumed to remain constant in test 1. In test 4, we deliberately force $\Theta_{surface}$ as described in the text of Sect. 2.1 to produce a warming surface. This forcing produces greater heat and moisture fluxes into the model domain from the surface, which then can affect the BL structure and cloud microphysics as described in Sect. 3.4. The w' $\Theta$' and w'$Q_{vap}$' fluxes shown in Figs. 3 and 10 show to what extent the surface plays a role in driving the BL convection development, with little surface input shown in Fig. 3 and significant contributions in Fig. 10. The presentation and discussion of these fluxes is as far as we take the evaluation of the input of the surface.

Specific Comments:

Title: "Large-scale: : : via : : :evaporation" and also enhanced cloud-top radiative cooling, right?

- Yes, this is correct. Reviewer 1 also raised this issue; therefore, we have simplified the title to: "*Relating large-scale subsidence to convection development in Arctic mixed-phase marine stratocumulus*". This therefore does not address specific reasons and creates a broader title to encapsulate all of the physical processes discussed.

Abstract Line 20: Clarify "warming surface", which you do not mean to be climatological, but rather southward advection.

- We have clarified this statement as requested. We have added more information closer to the beginning of the abstract when the sensitivity studies are first introduced (page 1, lines 7-8):

  *"... and a warming surface (representing motion southwards)..."*

Page7 Line5: "an" should be "a"

- Changed as requested

Page9 Line9: For the cases that become dynamically coupled, is the surface becoming a moisture source?

- Yes, this is what we believe is happening. From Fig. 3(h), the two subsidence cases (which become dynamically coupled) have a greater upward flux of moisture from the surface and lower BL than the CNTRL simulation. The

timestep shown is 9 h, but this effect is also seen at the later times (9 – 12 h) when these cases tend towards coupling.

 Why the later, more rapid increase in CNTRLD10x2? Is this important somehow to understand the main thesis of this simulation?

- The reason is not entirely clear; however, we suspect it is a similar manifestation of the ice phase influencing cloud dynamics as shown in the D10 ocean case in Young et al., 2017. In summary, the higher ice number concentration modelled in the D10x2 cases may be acting to introduce localised regions of convection into these mixed-phase clouds, with hot-spots of LWP forming. The processes involved in this hypothesis are summarised in Sect. 5.2 of Young et al., 2017.We have included a summary of this discussion in the manuscript at Page 11, lines 32-33:

  *"The cause of this increase is not clear; however, it may be due to increased localised cloud convection caused by the high $N_{ice}$, which has been previously modelled by Young et al., 2017."*

 "earlier" not "more quickly"?

- We agree with your suggestion and have made this change in the manuscript (now page 11, line 31).

 This doesn't seem right. Looks like the LWP response to Nice is much larger than the response to subsidence.

- This is correct and was an error on our part. This text has been changed to better describe Fig. 6 (page 12, lines 1-3). $N_{ice}$ does indeed affect the LWP more than $W_{sub}$; however, the addition of subsidence acts to stabilise the LWP timeseries, producing stable, or even increasing, trends. This stability would help to sustain the cloud against glaciation through the WBF mechanism, whereas the decreasing LWP trends of the CNTRL simulations would be susceptible to this phenomenon.

 Is the ascent of the cloud exacerbating the difference relative to CNRTL in 7B(a,d) since its spatial position is changing relative to CNTRL?

- Yes we do believe this will be a factor in the interpretation of the D10x0.5 simulations in Fig. 7B (and 7A). Subsidence acts to maintain cloud top height, as shown by the contoured number concentrations in Fig. 7A(a,d), B(a,d). Therefore, positive rain/snow production rates above approximately 1500m are not true "production": in the LO- and HISUB cases, the production rates at these altitudes are zero, whilst they are strongly negative in the ascending CNTRL. As a result, this manifests as a positive production rate in the subsidence cases. We have included discussion of this point to ensure the reader is not mislead (page 14, lines 1-5):

  *"Consequently, cloud top height increases in D10x0.5, while this ascent is suppressed in D10x2. This ascent adds complexity into the interpretation of Figs. 7A(a,d), B(a,d) as we are comparing clouds which are ascending at different rates. Strong cloud-top evaporation/sublimation of rain/snow is modelled above 1500m with the ascending CNTRL cloud, whilst the LO- and HISUB cases have no activity at these altitudes; therefore the anomaly between the $W_{sub}$ and CNTRL simulations appears positive at these heights."*

Page21 Line4: Replace "extent" with "depth" or "physical thickness" so as not to be confused with horizontal extent.

- This has been changed to "depth" as requested (page 19, line 6).

[revised manuscript text omitted]